# U-Cast: A Surprisingly Simple and Efficient
# Frontier Probabilistic AI Weather Forecaster

**Salva Rühling Cachay** [1]  **Duncan Watson-Parris** [1]  **Rose Yu** [1]

## Abstract

AI-based weather forecasting now rivals traditional physics-based ensembles, but state-of-the-art (SOTA) models rely on specialized architectures and massive computational budgets, creating a high barrier to entry. We demonstrate that such complexity is unnecessary for frontier performance. We introduce U-Cast, a probabilistic forecaster built on a standard U-Net backbone trained with a simple recipe: deterministic pre-training on Mean Absolute Error followed by short probabilistic fine-tuning on the Continuous Ranked Probability Score (CRPS) using Monte Carlo Dropout for stochasticity. As a result, our model matches or exceeds the probabilistic skill of GenCast and IFS ENS at $1.5°$ resolution while reducing training compute by over $10\times$ compared to leading CRPS-based models and inference latency by over $10\times$ compared to diffusion-based models. U-Cast trains in under 12 H200 GPU-days and generates a 15-day ensemble forecast in 3 seconds. These results suggest that scalable, general-purpose architectures paired with efficient training curricula can match complex domain-specific designs at a fraction of the cost, opening the training of frontier probabilistic weather models to the broader community. Our code is available at: https://github.com/Rose-STL-Lab/u-cast.

## 1. Introduction

The field of Artificial Intelligence-based Weather Prediction (AIWP) has rapidly evolved from a scientific curiosity to a viable alternative to operational Numerical Weather Prediction (NWP). While early deterministic models (Lam et al., 2023; Bi et al., 2023; Bodnar et al., 2024) demonstrated superior skill in forecasting mean atmospheric states,

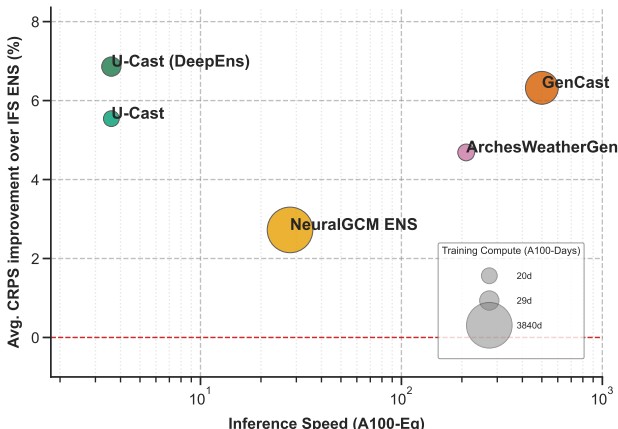

*Figure 1.* **The Efficiency-Accuracy Pareto Frontier.** We visualize forecast skill (y-axis, % improvement over IFS ENS), inference latency (x-axis), and training cost (bubble size). Our model (top-left) achieves state-of-the-art performance while requiring ***an order of magnitude less compute for training and/or inference*** compared to leading baselines. See Appendix C.1 for detailed methodology.

they fundamentally struggle with the chaotic nature of the atmosphere. Minimizing mean error forces these models to output the conditional mean of future states, resulting in "blurry" forecasts that lack physical realism at longer horizons (Brenowitz et al., 2025). Consequently, the field has shifted toward probabilistic ensembles—such as GenCast (Price et al., 2024) and FGN (Alet et al., 2025)—which characterize uncertainty and have recently surpassed the skill of the ECMWF ensemble (IFS ENS) (ECMWF, 2019), the gold standard in operational meteorology.

While these advances are remarkable, they have coincided with a dramatic increase in architectural complexity and computational cost, particularly when contrasted with the field's convolutional origins (Weyn et al., 2019; Rasp & Thuerey, 2021). Modern SOTA models employ complex designs, such as iterative diffusion processes (Price et al., 2024; Couairon et al., 2026), graph transformers (Price et al., 2024; Alet et al., 2025; Lang et al., 2024), or spherical neural operators (Bonev et al., 2025), often necessitating massive compute clusters. Training budgets routinely exceed hundreds of TPU- or GPU-days, even at coarser resolutions like $1°$ or $1.5°$. This trend raises practical barriers for reproducibility, iteration, and broader participation in frontier weather modeling (Bauer, 2024).

---

[1]UC San Diego. Correspondence to: Salva Rühling Cachay <salvaruehling@gmail.com>.

*Proceedings of the $43^{rd}$ International Conference on Machine Learning*, Seoul, South Korea. PMLR 306, 2026. Copyright 2026 by the author(s).

This trajectory raises a fundamental scientific question: **Is this complexity necessary?** Does frontier performance truly require modeling the Earth as a graph or training expensive generative ensembles from scratch? Or can a general, simple, and efficient design suffice if properly optimized (Qu & Krishnapriyan, 2024)?

In this work, we demonstrate that the latter is surprisingly true. We introduce U-Cast, a streamlined approach that prioritizes simplicity and extreme efficiency without compromising on forecast skill. Our methodology is defined by three key design choices. (1) We revert to a *standard U-Net backbone* with bottleneck self-attention, eschewing complex graph or spherical operations. This leverages the predominantly local nature of atmospheric dynamics at short time scales, which is well-suited to efficient convolutional architectures. (2) We introduce a *two-stage curriculum*: pre-train a deterministic forecaster using Mean Absolute Error (MAE) and fine-tune it probabilistically using the Continuous Ranked Probability Score (CRPS). This curriculum significantly accelerates convergence compared to training ensembles from scratch. (3) We simplify the source of stochasticity by using *Monte Carlo Dropout* (Gal & Ghahramani, 2016). Unlike the adaptive LayerNorm-based noise injection used in prior models (Lang et al., 2024; Alet et al., 2025), Dropout is parameter-free, reducing total parameter count by $\approx 10\%$ while maintaining or improving empirical performance. These choices are complemented by the *Muon optimizer* (Jordan et al., 2024), which we find offers superior generalization and convergence speed over AdamW in this domain (Section 4.4).

In summary, our contributions are as follows:

- **Frontier performance with simplicity:** Despite its simple design, our model achieves CRPS scores at $1.5°$ resolution competitive with GenCast, the leading public probabilistic weather forecasting model. U-Cast outperforms GenCast by up to 3% and IFS ENS by up to 23.7% in CRPS on specific variables and lead times, such as short-range $500\,\text{hPa}$ geopotential and mean sea level pressure, respectively.

- **A recipe for efficient probabilistic training:** We demonstrate that the high computational cost of ensemble training is not inherent to the problem but a byproduct of inefficient training pipelines. By decoupling the learning of physics (deterministic pre-training) from the learning of uncertainty (probabilistic fine-tuning), paired with simple Dropout-based stochasticity and effective optimization, we reduce the cost of training a frontier generative model by an order of magnitude.

- **Extreme efficiency:** As shown in Figure 1, our model occupies a distinct position on the Pareto frontier. It trains in under 3 days on 4 H200 GPUs and generates a single 60-step forecast in only 2 seconds. This represents an order-of-magnitude reduction in training and/or inference cost compared to SOTA baselines.

## 2. Related Work

**Architectural backbones & the complexity trap.** Regardless of whether the training objective is deterministic (MSE) or probabilistic, the field has moved towards specialized architectures to manage the spherical geometry of the Earth. Graph networks, pioneered by Keisler (2022), serve as the backbone for the deterministic GraphCast (Lam et al., 2023) as well as the probabilistic GenCast (Price et al., 2024), FGN (Alet et al., 2025), AIFS-CRPS (Lang et al., 2024), and Graph-EFM (Oskarsson et al., 2024). Fourier or spherical neural operators, which enforce geometric symmetries, are used in the deterministic FourCastNet (Pathak et al., 2022) and its ensemble-forecasting successors (Cachay et al., 2024; Bonev et al., 2025; Mahesh et al., 2025). Similarly, vision transformers—often adapted using 3D-Swin blocks or custom variable tokenization—underpin models such as Pangu (Bi et al., 2023), Aurora (Bodnar et al., 2024), Stormer (Nguyen et al., 2024), FuXi (Chen et al., 2023), and others (Zhong et al., 2025; Stock et al., 2025; Hatanpää et al., 2025). While elegant, these designs introduce significant engineering complexity. Convolutional architectures have been explored (Weyn et al., 2019; Karlbauer et al., 2024; Cresswell-Clay et al., 2025; Andrae et al., 2025), but have received less attention in recent frontier work on medium-range weather forecasting. We revisit this direction, demonstrating that the "inductive bias" of geometric architectures is less critical than previously thought, even for frontier probabilistic forecasting.

**The efficiency trade-off in probabilistic AIWP.** To resolve the "blurriness" of deterministic models, the field has bifurcated into two high-cost directions. Diffusion and flow-matching models, such as GenCast (Price et al., 2024) and others (Couairon et al., 2026; Cachay et al., 2025; Nguyen et al., 2025), generate highly realistic ensembles but incur significant *inference costs* due to iterative denoising (requiring dozens of forward passes per forecast). Conversely, CRPS-optimized ensembles like AIFS-CRPS (Lang et al., 2024), FGN (Alet et al., 2025), and FourCastNet3 (Bonev et al., 2025) offer fast inference but incur massive *training costs*, as they must generate full ensemble forecasts during training to compute the loss. Alternatively, hybrid approaches like NeuralGCM (Kochkov et al., 2024) couple a differentiable dynamical core with ML components but face challenges in spatial scalability. This creates a dilemma: frontier methods typically impose considerable computational burdens during training, inference, or both.

**Our approach: U-Cast.** We introduce a third path that avoids this trade-off. We synthesize a minimally-adapted

off-the-shelf U-Net backbone with a streamlined probabilistic training recipe. Unlike diffusion models, U-Cast requires only a single forward pass *per member* (fast inference). Unlike standard CRPS ensembles, our deterministic-to-probabilistic curriculum avoids the need for expensive ensemble training from scratch (fast training) and uses simple MC dropout instead of noise injection for stochasticity.

**Concurrent work.** Concurrent work has explored CRPS-based fine-tuning of deterministic backbones (Schreck et al., 2025; Diaconu et al., 2026). However, these approaches retain the noise injection modules of prior CRPS-trained models and require considerably longer probabilistic fine-tuning ($> 10\times$ more gradient steps) than our recipe. Zhdanov et al. (2026) introduce MOSAIC, a $1.5°$ probabilistic forecaster that combines CRPS training and noise injection with an efficient mesh-aligned block-sparse attention mechanism over the HEALPix grid, achieving strong spectral fidelity at low training and inference cost (Table 2).

# 3. Methodology

We frame probabilistic weather forecasting as learning a probabilistic mapping from the current atmospheric state to future states. Let $\boldsymbol{x}_t \in \mathbb{R}^{C \times H \times W}$ represent the state of the atmosphere at time $t$, where $C$ denotes the number of variables (e.g., geopotential, humidity) and $H \times W$ represents the spatial grid resolution (e.g., $121 \times 240$ for $1.5°$).

Given two past states $\boldsymbol{x}_{t-1:t} = (\boldsymbol{x}_{t-1}, \boldsymbol{x}_t)$, we aim to model the conditional probability distribution of the next state, $p(\boldsymbol{x}_{t+1}|\boldsymbol{x}_{t-1:t})$. At inference, the model is rolled out autoregressively to generate trajectories for arbitrary lead times. We assess the quality of these probabilistic forecasts using the Continuous Ranked Probability Score (CRPS) (Matheson & Winkler, 1976), a standard proper scoring rule for ensembles (Hersbach, 2000; Rasp et al., 2024).

To achieve state-of-the-art skill efficiently, we propose **U-Cast**, a simple framework defined by three core pillars: a minimally-adapted standard U-Net backbone, a two-stage training curriculum, and MC Dropout-based stochasticity.

## 3.1. Architecture: Revisiting the U-Net

In contrast to prior works using graph (Lam et al., 2023; Price et al., 2024; Lang et al., 2024; Alet et al., 2025) or 3D Swin transformers (Bi et al., 2023; Bodnar et al., 2024; Couairon et al., 2026), and spherical neural operators (Bonev et al., 2023; Mahesh et al., 2025; Bonev et al., 2025), we employ a U-Net backbone widely adopted in image diffusion (Dhariwal & Nichol, 2021; Karras et al., 2022). While variants of this "DhariwalUnet" backbone have been explored—such as the 3D temporal version of Cachay et al. (2025) or the small-scale 3.5M-parameter version in Andrae et al. (2025)—we are the first to demonstrate that it

can achieve frontier probabilistic forecasting results. The strong local inductive bias of convolutions captures physical transitions effectively, while self-attention in the bottleneck layers resolves non-local interactions.

We make four modifications to the standard architecture, which center around scaling the model capacity, basic adaptations to better adhere to the data, and stripping away unnecessary modules: (1) we increase the model width to 320 channels in the initial layers, yielding 895M parameters to provide sufficient capacity for the high-dimensional atmospheric state space; (2) we use circularly padded convolutions along the longitude dimension to respect the Earth's periodic topology; (3) we add automatic bilinear upsampling in the decoder to match skip-connection dimensions when input grids are not strict powers of 2 (e.g., $121 \times 240$); and (4) we remove the adaptive LayerNorm (adaLN) conditioning layers present in the original DhariwalUnet, which are designed for diffusion timestep conditioning and are unnecessary in our non-diffusion framework without AdaLN-based noise injection. As discussed in Section 3.3, we instead rely on MC Dropout for stochasticity, which reduces the total parameter count by 5–10%.

Notably, this architecture is implemented in less than 300 lines of code. Compared to the $> 3000$ lines required for graph networks (Lam et al., 2023; Price et al., 2024) or approaches relying on custom spherical convolution packages (Bonev et al., 2025), our simple approach reduces the burden for maintenance and reproducibility.

## 3.2. Curriculum Learning: From MAE to CRPS

Training probabilistic models end-to-end on CRPS is computationally expensive. Generating multiple ensemble members ($M \geq 2$) per gradient step scales compute and memory costs linearly with $M$, which drives the massive training budgets associated with previous CRPS-based models (Alet et al., 2025; Lang et al., 2024; Bonev et al., 2025; Stock et al., 2025). To reduce the computational cost, we introduce a two-stage training curriculum that decouples the learning of atmospheric dynamics from the learning of forecast uncertainty. To unify notation, we define the pointwise $L_1$ distance (absolute error) between state vectors $\mathbf{u}$ and $\mathbf{v}$ at grid location $(h, w)$ as:

$$\delta_{h,w}(\mathbf{u}, \mathbf{v}) = \|\mathbf{u}_{h,w} - \mathbf{v}_{h,w}\|_1 \tag{1}$$

**Stage 1: Deterministic Pre-training.** First, we train the U-Net, $f_\theta$, to approximate the conditional mean of the future state. We minimize the spatially weighted MAE:

$$\mathcal{L}_{\text{det}} = \frac{1}{HW} \sum_{h,w} a_h \delta_{h,w}(f_\theta(\boldsymbol{x}_{t-1:t}), \boldsymbol{x}_{t+1}) \tag{2}$$

where $a_h$ represents latitude-dependent area weights (Rasp et al., 2024). We deliberately select the MAE rather than

the more common MSE because for a single deterministic forecast MAE equals the CRPS, ensuring that the loss landscape of Stage 1 is smoothly aligned with the probabilistic objective in Stage 2. Because this stage requires only a single forward pass, we can afford to train for an extended duration (100 epochs) at low cost. This allows the model to robustly learn fundamental atmospheric dynamics before transitioning to the expensive probabilistic phase.

**Stage 2: Probabilistic Fine-tuning.** Once the backbone has converged on the dynamics, we switch to probabilistic training to learn forecast uncertainty. We enable stochasticity via MC Dropout masks $\xi^{(m)}$, generating an ensemble of $M$ forecasts $\hat{\boldsymbol{x}}^{(m)} = f_\theta(\boldsymbol{x}_{t-1:t}; \xi^{(m)})$. We optimize the unbiased CRPS estimator (Zamo & Naveau, 2018). We calculate the "Skill" (ensemble MAE) and "Spread" (ensemble diversity) terms for each grid point locally:

$$\text{Skill}_{h,w} = \frac{1}{M} \sum_{m=1}^{M} \delta_{h,w}(\hat{\boldsymbol{x}}^{(m)}, \boldsymbol{x}_{t+1}) \quad (3)$$

$$\text{Spread}_{h,w} = \frac{1}{M(M-1)} \sum_{m=1}^{M} \sum_{n \neq m}^{M} \delta_{h,w}(\hat{\boldsymbol{x}}^{(m)}, \hat{\boldsymbol{x}}^{(n)}) \quad (4)$$

The final loss is the spatially weighted CRPS average:

$$\mathcal{L}_{\text{prob}} = \frac{1}{HW} \sum_{h,w}^{HW} a_h \left( \text{Skill}_{h,w} - \frac{1}{2}\text{Spread}_{h,w} \right) \quad (5)$$

Following Alet et al. (2025), we use the minimal ensemble size of $M = 2$ during training. Because the backbone has already learned the atmospheric dynamics in Stage 1, we find that this stage converges in only 8 epochs. As a result, despite the doubled per-step cost ($M = 2$), probabilistic fine-tuning accounts for only 15% of the total training budget. This strategy effectively amortizes the cost of learning physical dynamics, enabling rapid iteration on probabilistic design choices and the efficient training of deep ensembles.

**Stage 3: Deep ensembling.** To further improve probabilistic skill, we construct a deep ensemble (Lakshminarayanan et al., 2017) by repeating Stage 2 $K$ times with independent random seeds, each starting from the *same* Stage 1 deterministic checkpoint. At inference, each of the $K$ fine-tuned checkpoints generates $N$ stochastic rollouts via MC Dropout, yielding a combined ensemble of $KN$ members. In our experiments, we use $K = 4$ as in (Couairon et al., 2026; Alet et al., 2025). This design is uniquely enabled by the extreme training efficiency of U-Cast's probabilistic fine-tuning stage. Each additional deep ensemble member requires only the cost of Stage 2 (1.2 H200-days), which is two orders of magnitudes less compute per deep ensemble model than in FGN (Alet et al., 2025). we refer to the final model as U-Cast DeepEns (DE), while U-Cast refers to a single Stage 2 fine-tuned model without deep ensembling.

### 3.3. Stochasticity via Monte Carlo Dropout

A critical component of probabilistic training is the mechanism for injecting stochasticity to generate ensemble members, $\hat{\boldsymbol{x}}^{(m)}$. Prior literature on CRPS-training (Alet et al., 2025; Lang et al., 2024; Bonev et al., 2025; Stock et al., 2025) employs a noise injection strategy where random vectors modulate intermediate features via adaptive Layer-Norms. This specific choice may be derived from common practice in diffusion models (Karras et al., 2022; Price et al., 2024; Couairon et al., 2026) where it is used to condition on the diffusion step. While effective, this approach introduces additional parameters and has not been rigorously ablated.

We propose a simpler alternative: Monte Carlo Dropout (Gal & Ghahramani, 2016). Instead of specialized noise injection layers, we enable standard Dropout layers during both training and inference. In this framework, the stochastic variable $\xi$ introduced in Section 3.2 corresponds to a sampled Dropout mask. While early works applied Dropout to weather forecasting (Scher & Messori, 2021; Garg et al., 2022; Hu et al., 2023; Chen et al., 2023), they reported unsatisfying, under-dispersive ensembles. This tendency for Dropout to yield overconfident predictions has been observed in the broader machine learning literature as well (Lakshminarayanan et al., 2017; Huang et al., 2023; Sun & Yu, 2024).

We show that this issue is not intrinsic to Dropout, but rather a consequence of training objectives (such as MSE) that do not explicitly reward ensemble spread. When coupled with the CRPS objective of Section 3.2, the model learns to leverage dropout masks—equivalently, sampled sub-networks—to produce relatively well-calibrated probabilistic forecasts. Our approach offers three advantages over noise injection: (1) Dropout applies seamlessly across both training stages. It acts as a standard regularizer during deterministic pre-training and transitions into an ensemble generator during CRPS fine-tuning, avoiding the introduction of new, untrained parameters; (2) removing the noise projection and adaptive LayerNorm layers reduces the total parameter count by 5–10%; and (3) as detailed in Section 4.4, this simpler mechanism yields better probabilistic skill in terms of CRPS than noise injection.

## 4. Experiments

### 4.1. Experimental Setup

**Data.** We benchmark our model on medium-range weather forecasting using the ERA5 reanalysis dataset (Hersbach et al., 2020). We use the $1.5°$ resolution of the data provided by (Rasp et al., 2024). Following GenCast and FGN, we use six atmospheric variables at 13 pressure levels, as well as five surface variables as inputs and outputs, complemented by two static

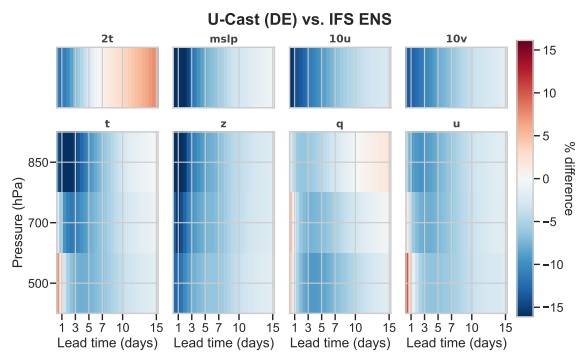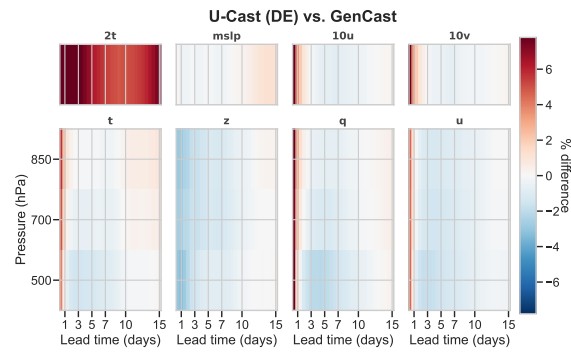

*Figure 2.* $1.5°$ **CRPS comparison of U-Cast DeepEns against IFS ENS (left) and GenCast (right).** Blue indicates lower (better) CRPS for U-Cast; red indicates baseline superiority. U-Cast broadly outperforms IFS ENS and is competitive with GenCast despite the latter's finer native resolution $(0.25°)$. See text for details.

and four time forcings as conditioning signals. In total, this results in 83 output features and $172 = 83 \times 2 + 6$ input features (since we use two past timesteps as inputs), see Table 3 for a full description of the variables. We train our models on data from 1979 to 2019. Following GenCast, we train our model on 12-hourly data, i.e., sequences of `00z/12z` or `06z/18z`, using a window of two past input snapshots and predicting the residual with respect to the last input snapshot.

**Architecture details.** We use 4 down- and up-sampling blocks each (channel multipliers: 1, 2, 3, 4; base width: 320, 4 residual convolutional layers per resolution), with attention restricted to the two coarsest spatial resolutions. We apply 10% dropout and perform inference using an Exponential Moving Average (EMA) of model weights (decay 0.9999).

**Optimizer.** We depart from the AdamW optimizer used ubiquitously in AIWP and employ the Muon optimizer (Jordan et al., 2024), a momentum-orthogonalized method originally designed for large-scale language and vision training. As shown in our ablations (Section 4.4), Muon not only accelerates convergence but also finds better optima, yielding lower final CRPS than AdamW—particularly during the probabilistic fine-tuning stage, where we found the optimizer choice to be most critical to achieve the extreme efficiency of our training curriculum.

**Hyperparameters.** Both stages use an effective batch size of 48 and a linear warmup of 1500 steps. Since Muon cannot optimize 1D parameters (e.g., biases), we use AdamW for that subset throughout. *Stage 1* (deterministic, 100 epochs): cosine decay with peak learning rates of 3e-3 (Muon) and 3e-4 (AdamW), and weight decay of 0.1 and 0.03, respectively. *Stage 2* (probabilistic, 8 epochs): same warmup, decayed over the shorter schedule, with learning rates of 7e-3 (Muon) and 7e-5 (AdamW). We use the same variable-specific loss weights as in (Price et al., 2024).

### 4.2. Weatherbench 2 $1.5°$ Evaluation

*Table 1.* Absolute CRPS results (lower is better) for geopotential at 500hPa and 10m u-component of wind for 1, 3, and 10-day ensemble forecasts. See Figure 3 for more comprehensive results.

| Model | z500 | | | 10u | | |
|---|---|---|---|---|---|---|
| | 1d | 3d | 10d | 1d | 3d | 10d |
| U-Cast | 20.3 | 55.2 | 256 | 0.349 | 0.61 | 1.55 |
| U-Cast (DE) | **19.6** | **53.5** | **253** | 0.345 | **0.60** | **1.54** |
| IFS ENS | 22.4 | 58.3 | 262 | 0.406 | 0.69 | 1.61 |
| GenCast | 20.2 | 54.3 | 254 | **0.332** | **0.60** | 1.55 |
| ArchesGen | 21.2 | 55.8 | 254 | 0.370 | 0.62 | 1.55 |
| NeuralGCM | 22.9 | 54.6 | 254 | - | - | - |

To situate U-Cast within the broader landscape of probabilistic weather forecasting, we use the established WeatherBench 2 benchmark (Rasp et al., 2024). We run inference on all 732 initial conditions (`00z` and `12z`) from 2020.

An important caveat for interpreting these results: the GenCast scores on the $1.5°$ WeatherBench 2 leaderboard are derived from regridding its native $0.25°$ forecasts—a finer resolution than our native grid. This confers a systematic advantage, as the higher-resolution inputs can resolve surface heterogeneity crucial for variables such as 2-meter temperature. With this asymmetry in mind, we consider the comparison conservative for U-Cast.

In Figure 2, we visualize the CRPS of U-Cast's deep ensemble (DE) relative to the leading physics-based model, IFS ENS, and the leading open-source AI model, GenCast. U-Cast (DE) improves upon IFS ENS on 92.9% of variable–lead time combinations, with an average CRPS gain of 5.0% and peak improvements of 23.7% for short-range mean sea level pressure. Despite the resolution disadvantage noted above, U-Cast (DE) achieves an average improvement of 0.21% over GenCast, with gains of up to 3% for short-range `z500`. The primary deficits are concentrated in 2-meter temperature and 12-hour lead times—precisely the settings

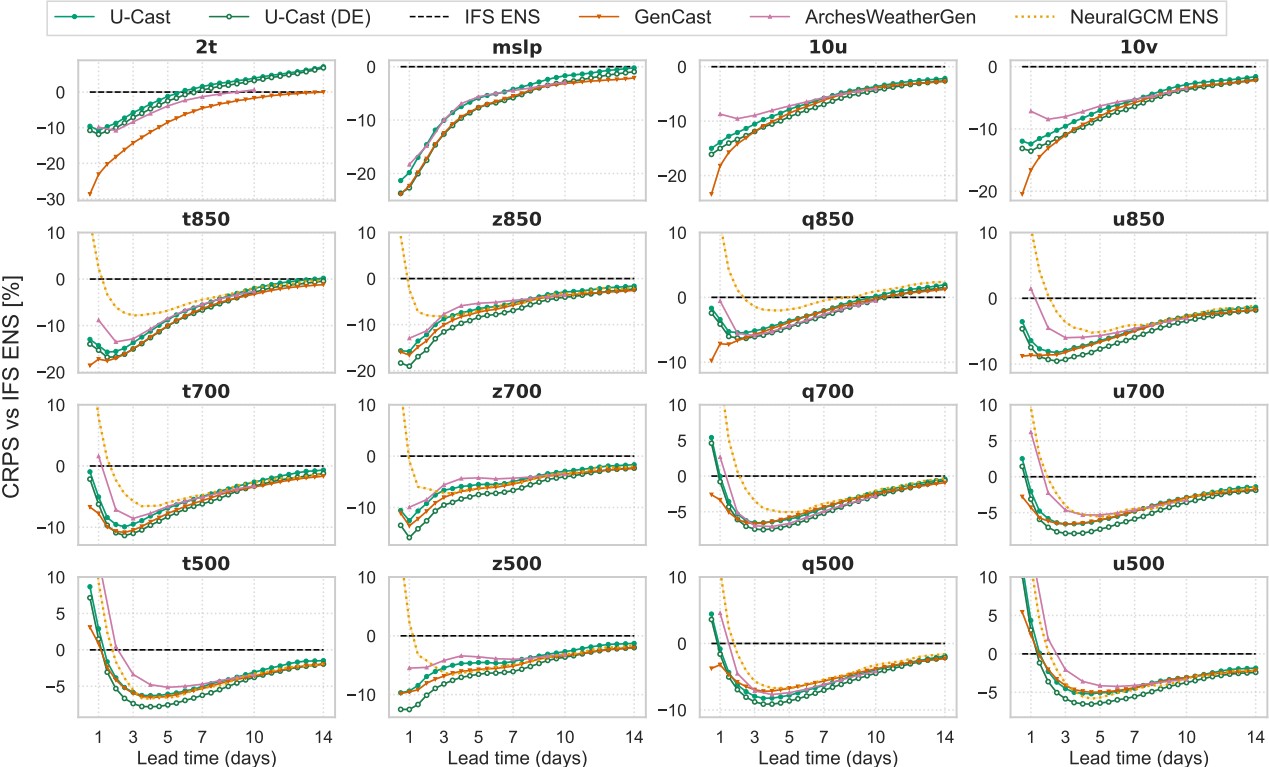

*Figure 3.* **WeatherBench 2 Comparison** ($1.5°$ **resolution**). We report the CRPS skill relative to the operational IFS ENS (%, lower is better) as a function of forecast horizon. Baseline scores are sourced from the official leaderboard (Rasp et al., 2024). Numbers after variable abbreviations denote pressure levels in hPa. Note that the GenCast baseline is the native $0.25°$ model regridded to $1.5°$ (see Section 4.2 for discussion of this resolution asymmetry). U-Cast consistently outperforms baselines on geopotential variables while remaining highly competitive on other fields.

where GenCast's finer native resolution would be most beneficial. Figure 3 and Table 1 provide a broader comparison against additional baselines across variables and lead times.

Extended analysis in Appendix C.4 confirms that this strong performance persists when benchmarking ensemble-mean RMSE (Figure 8) and when evaluating on 2022 instead of 2020 (Figure 10), where performance degrades by less than 4.5% despite U-Cast being trained only on 1979–2019 data. Additionally, we provide a qualitative analysis of forecast realism in Appendix C.8, complemented by a power spectra analysis (Section C.7), which identifies systematic artifacts in the polar regions. This limitation could be due to the 2D U-Net architecture failing to capture the spherical topology of the data, which could be addressed with a lightweight spherical variant of our U-Net (Esteves et al., 2023; Karlbauer et al., 2024).

Analyzing the spread-skill ratio (Figure 9) reveals that U-Cast produces slightly under-dispersive forecasts at short-to-medium horizons—though notably less so than prior MC Dropout-based weather models (Scher & Messori, 2021; Garg et al., 2022; Chen et al., 2023). We attribute this to parameter sharing across dropout masks, which induces

correlated ensemble members; consistently, our adaLN ablation (Section 4.4) yields better dispersion but worse CRPS. Our deep ensemble variant mitigates this, achieving SSR above 0.85 across nearly all variables and lead times and outperforming NeuralGCM ENS in short-range calibration (Figure 9). Further gains could likely be obtained via initial-condition perturbations (Leutbecher & Palmer, 2008). Overall, U-Cast establishes a new efficiency frontier, matching or exceeding the skill of computationally intensive baselines while operating with training and/or inference budgets that are orders of magnitude smaller (cf. Section 4.3).

### 4.3. Computational Costs

Table 2 benchmarks the computational resources required for U-Cast against probabilistic baselines. At comparable resolutions (both $1°$ and $1.5°$), our approach establishes a distinct efficiency advantage. While comparable $1°$ CRPS-optimized baselines such as FGN (Alet et al., 2025) and AIFS-CRPS (Lang et al., 2024) require 300 TPU-days and 256 H100-days, respectively, to converge, U-Cast requires only 15 H200-days. This corresponds to a reduction in training cost of more than one order of magnitude. Only ArchesWeatherGen (Couairon et al., 2026) and ERDM (Cachay

*Table 2.* Training compute costs and inference speeds for probabilistic weather forecasting models. Models are grouped by resolution. Inference speed is for one single-member 60-step rollout (15-days for NeuralGCM and IFS ENS, or if a 6h-resolution model). [†] TPU benchmarks for $1°$ GenCast unavailable, but official documentation indicates a $\approx 3\times$ speed-up over a H100 GPU at $0.25°$. [*] For each of the 4 individual models. $1°$ cost is estimated.

| Model | Training Cost (in days) | Inference Speed |
|---|---|---|
| *Coarser Resolution* ($\sim 1.5°$) | | |
| ArchesWeatherGen | 45 (V100) | 3.5 min (A100) |
| NeuralGCM ENS | 1280 (TPUv5e) | 18.6 sec (TPUv4) |
| ERDM | 20 (H200) | 7 min (A100) |
| Swift | 360 (Intel Max1550) | – |
| MOSAIC | 16 (H100) | 3 sec (H100) |
| **U-Cast (ours)** | **8.2 (11.8 for DE**; H200) | **2/3.6 sec (H100/A100)** |
| *Mid-resolution* ($\sim 1°$) | | |
| GenCast (Stage 1) | 112 (TPUv5) | $\sim$5 min (H200)[†] |
| FGN (Stage 1) | 300 (TPUv5p/6e)[*] | – |
| AIFS-CRPS O96 | 256 (H100) | $\sim$1 min (A100) |
| **U-Cast (ours)** | **15 (H200)** | **3/5.2 sec (H100/A100)** |
| *High Resolution* ($\sim 0.25°$) | | |
| GenCast (+Stage 2) | 160 (TPUv5) | $\sim$ 16 min (TPUv5) |
| FGN (+Stage 2) | 490 (TPUv5p/6e)[*] | $\sim$ 1 min (TPUv5p) |
| FourCastNet-3 | 3328 (H100) | 1 min (H100) |
| AIFS-CRPS N320 | 896 (H100) | $\sim$4 min (A100) |
| IFS ENS (at $\sim 0.1°$) | N/A (Not AI) | 1hr (96 AMD CPUs) |

et al., 2025) report a similarly low training burden (albeit at the coarser $1.5°$ resolution); however, their reliance on iterative diffusion sampling incurs $> 10\times$ higher inference cost than our single-pass approach. While diffusion inference latencies remain small compared to operational NWP (e.g., IFS ENS), U-Cast's extreme inference speed opens the door to data-intensive applications previously constrained by compute, such as routinely generating thousand-member ensembles for robust extreme-event detection (Mahesh et al., 2025; Deser et al., 2020; McKinnon & Simpson, 2022). The inference times reported in Table 2 correspond to a *single ensemble member*; batching members scales sublinearly. For example, U-Cast completes a 60-step rollout (a 30-day horizon at 12-hour resolution) on an H100 in 2 seconds for a single member versus 12 seconds for ten. Further details on these estimates are provided in Appendix C.2.

### 4.4. Ablations

In Figure 4, we ablate key design choices.

**Optimizer.** We retrain U-Cast using AdamW instead of Muon, applying a learning rate of 3e-4 for pre-training and sweeping $\{3e\text{-}5, 1e\text{-}4, 3e\text{-}4\}$ for fine-tuning. This substitution causes the most significant performance drop, degrading `z500` CRPS by up to 15% in the 1-to-3-day range. We found Muon to be particularly critical for the probabilistic fine-tuning stage; while it accelerates convergence during deterministic pre-training, AdamW can eventually reach similar optima given sufficient time. This result underscores that the Muon advantage is primarily one of efficiency: it

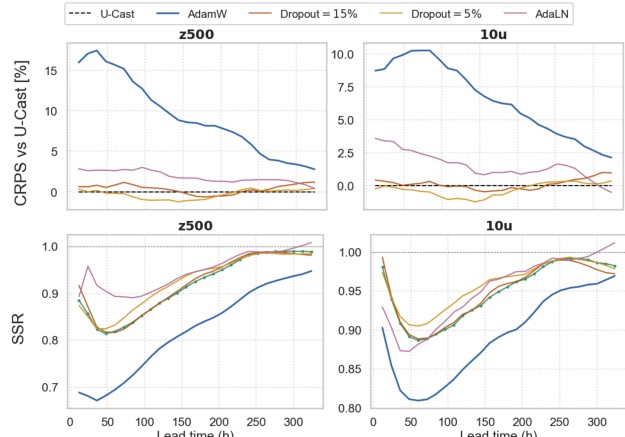

*Figure 4.* **U-Cast ablations.** We report CRPS relative to U-Cast (top row) and spread-skill ratio (bottom row; closer to 1 is better).

enables the curriculum's brief fine-tuning stage (8 epochs) to converge to frontier-quality probabilistic forecasts. An AdamW-based pipeline could plausibly reach similar skill, but would require substantially longer CRPS training or even CRPS training from scratch, which would negate the computational savings that are central to our approach.

**Stochasticity Source.** We investigate the sensitivity of our model to the stochastic mechanism by varying the Monte Carlo dropout rate (5%, 15%) and testing noise injection via adaptive LayerNorm (adaLN) (Lang et al., 2024; Alet et al., 2025) as an alternative. For these experiments, we freeze the pre-trained backbone and only re-run the probabilistic fine-tuning stage. The adaLN variant projects a 32-dimensional noise vector to scale and shift feature maps; we zero-initialize the projection layer to mitigate training shock. Results indicate that while adaLN generally improves dispersion, it significantly degrades CRPS for 1-to-7-day forecasts. Regarding dropout, performance is robust, with all rates falling within $\pm 1\%$ CRPS of the baseline. Counter-intuitively, the lowest dropout rate (5%) yields the best calibration (SSR closest to 1), reducing the model's tendency toward under-dispersion compared to higher rates. We also find that increasing the training ensemble size from $M = 2$ to $M = 4$ yields only marginal CRPS gains despite doubling per-step cost (Figure 12), validating our choice of the minimal ensemble size.

**Curriculum.** We validate our two-stage curriculum by training U-Cast on the CRPS from scratch—without deterministic pre-training—using the same total compute budget (8.2 H200-days, or 50 epochs of end-to-end CRPS training). As shown in Figure 5, the curriculum-trained model converges to a validation CRPS of 0.218 for `t850` at 12h lead time, a 3.4% improvement over the from-scratch baseline (0.225), reaching this score in under 15k gradient steps ver-

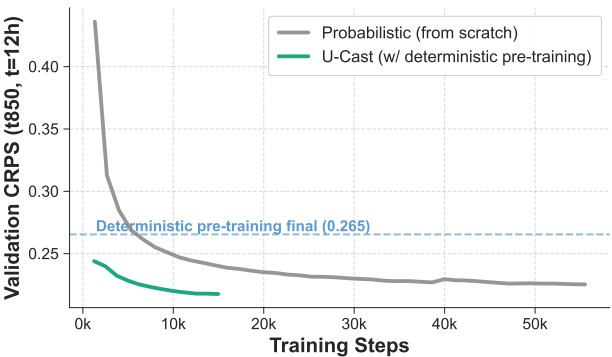

*Figure 5.* **Curriculum ablation.** Validation CRPS (`t850`, 12h lead time) during probabilistic training. The curriculum (orange) fine-tunes from a deterministic checkpoint (dashed blue line) and converges rapidly to a CRPS of 0.218. Training from scratch on CRPS alone (gray) requires $> 3\times$ more steps to reach comparable performance and plateaus at a worse score (0.225).

sus over 50k for the baseline. Full evaluation across more variables and lead times (Figure 12) shows that the from-scratch baseline consistently degrades short-range CRPS by 3–5% for most surface and upper-air variables and 5–15% for stratospheric variables, though it occasionally recovers long-range scores for select variables such as 2t. On balance, the curriculum dominates while using a fraction of the probabilistic training budget, validating the core premise of our recipe: decoupling the learning of physics from uncertainty enables both training efficiency and forecast quality.

## 5. Discussion

### 5.1. Escaping the Complexity Trap

The recent history of AI weather prediction has been characterized by rapidly increasing costs in architecture, inference, and training. U-Cast addresses each on three distinct fronts:

**Architectural simplicity.** Leading approaches employ specialized geometric designs (Price et al., 2024; Bonev et al., 2025) and elaborate stochasticity mechanisms (Price et al., 2024; Lang et al., 2024; Alet et al., 2025). U-Cast's competitive performance with a standard U-Net and MC Dropout suggests that both geometric inductive biases and potent noise-injection schemes are less critical for medium-range forecasting than previously assumed (with caveats discussed in Section 6).

**Inference efficiency (CRPS vs. diffusion).** Like other CRPS-based models (Alet et al., 2025; Lang et al., 2024), U-Cast avoids the iterative denoising cost of diffusion (Price et al., 2024; Couairon et al., 2026; Cachay et al., 2025), reducing inference latency by an order of magnitude while matching probabilistic skill. Crucially, unlike prior CRPS models, U-Cast achieves this without incurring a prohibitive training cost (see next point).

**Training efficiency (curriculum & Muon).** Optimizing CRPS requires generating full ensembles at every training step, making prior CRPS models prohibitively expensive (Alet et al., 2025; Lang et al., 2024; Bonev et al., 2025; Stock et al., 2025). Our curriculum decouples learning atmospheric dynamics from learning forecast uncertainty: the expensive probabilistic stage accounts for only 15% of total compute, and Muon ensures that this short stage actually converges to a strong optimum. This decoupling also enables efficient deep ensembling, as only the lightweight fine-tuning stage needs to be repeated per member.

### 5.2. Democratizing the AIWP Frontier

One of our most significant contributions is lowering the barrier to entry for AI weather research. Currently, training frontier probabilistic weather models is restricted to large industrial or national labs with access to massive clusters. The trend toward increasingly complex, computationally intensive architectures concentrates frontier development in a small number of well-resourced labs. In contrast, our complete recipe trains in 3 days on 4 H200 GPUs—or, if using a pre-trained deterministic backbone, fine-tuned probabilistically in around 1 day on a single H200 GPU. This order-of-magnitude efficiency gain (Figure 1) empowers academic labs and under-resourced practitioners to not merely consume, but actively contribute to the development of frontier weather models.

## 6. Conclusion

U-Cast provides an existence proof that frontier probabilistic weather forecasting can be achieved with standard deep learning components—a U-Net, Dropout, and a two-stage curriculum effectively optimized with Muon—at a fraction of the cost of current SOTA systems. We do not claim that domain-specific design is obsolete: our success still relies on a domain-aware objective (CRPS), and our analysis of polar artifacts highlights regimes where geometric inductive biases remain valuable. Rather, we advocate for a *complexity-by-necessity* approach: start with efficient, simple methods and incorporate specialized components only where they demonstrably add value.

**Limitations and future work.** Our approach is not without limitations. First, our power spectra and qualitative analyses reveal systematic artifacts (Appendix C.7 & C.8), likely stemming from our minimalist treatment of the sphere. This suggests that while geometric priors may be redundant for aggregate global forecasting skill, they remain valuable for topological edge cases, which could be addressed by operating the U-Net on a more suitable grid (Esteves et al., 2023; Karlbauer et al., 2024). Second, our model often exhibits slight under-dispersion, and rollouts beyond the medium

range (>20 days) become unstable. The instability might stem from the absence of autoregressive training, which prior work uses to improve long rollouts; extending the curriculum with an autoregressive fine-tuning stage is a natural next step. The remaining under-dispersion could be further reduced via initial-condition perturbations (Leutbecher & Palmer, 2008). Importantly, with sufficient compute, U-Cast could be scaled and fine-tuned to higher $0.25°$ resolution.

## Impact Statement

This paper presents an advancement in AI Weather Prediction (AIWP) that emphasizes efficiency and accessibility. The primary impact of this work is the *democratization of weather AI research.* By reducing the computational requirements for training frontier probabilistic models by more than $10\times$, we take a step towards enabling universities, researchers in the Global South, and smaller organizations to develop and adapt weather models to their local needs without requiring prohibitive capital investment. This is particularly enabled by the fact that our probabilistic fine-tuning recipe takes only 15% of the total compute budget, opening a pathway to rapidly build frontier probabilistic weather forecasting when a solid pre-trained deterministic forecaster already exists. This also translates to a lower carbon footprint for model development and deployment.

However, as with all data-driven forecasting systems, there are risks associated with out-of-distribution data. While our model provides uncertainty estimates, it does not encode explicit physical laws and cannot be expected to perform well on different data distributions than it was trained on (e.g., future climates). Users should exercise caution and, depending on their use-case, more comprehensively evaluate such models before trusting them.

## Acknowledgements

We acknowledge generous support from Modal Labs through a computational credit research grant. This research used resources from the National Energy Research Scientific Computing Center (NERSC), a Department of Energy User Facility, using NERSC awards DDR-ERCAP0036994 and FES-ERCAP0036998. This work was supported in part by the U.S. Army Research Office under Army-ECASE award W911NF-07-R-0003-03, the U.S. Department Of Energy, Office of Science, IARPA HAYSTAC Program, and NSF Grants #2205093, #2146343, #2134274, #2441832, CDC-RFA-FT-23-0069, DARPA AIE FoundSci and DARPA YFA.

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

# Appendix

## A. Dataset Details

In Table 3 we enumerate all the variables that our model ingests and predicts, largely following the setup of GenCast (Price et al., 2024) and FGN (Alet et al., 2025) with the exception of dropping the total precipitation variable from the set of predicted variables. The atmospheric variables are used at the 13 standard pressure levels from the Weatherbench 2 (Rasp et al., 2024) benchmark: 50, 100, 150, 200, 250, 300, 400, 500, 600, 700, 850, 925, and 1000 hPa.

*Table 3.* Description of data variables used.

| Type | Variable name | Short name | Role |
|------|---------------|------------|------|
| Atmospheric | Geopotential | z | Input/Predicted |
| Atmospheric | Specific humidity | q | Input/Predicted |
| Atmospheric | Temperature | t | Input/Predicted |
| Atmospheric | U component of wind | u | Input/Predicted |
| Atmospheric | V component of wind | v | Input/Predicted |
| Atmospheric | Vertical velocity | w | Input/Predicted |
| Single | 2 metre temperature | 2t | Input/Predicted |
| Single | 10 metre u wind component | 10u | Input/Predicted |
| Single | 10 metre v wind component | 10v | Input/Predicted |
| Single | Mean sea level pressure | mslp | Input/Predicted |
| Single | Sea Surface Temperature | sst | Input/Predicted |
| Static | Geopotential at surface | z | Input |
| Static | Land-sea mask | lsm | Input |
| Clock | Local time of day | n/a | Input |
| Clock | Elapsed year progress | n/a | Input |

## B. Metrics

Let $\boldsymbol{x} \in \mathbb{R}^{H \times W}$ denote the ground truth target field for a specific variable and time step, and $\hat{\boldsymbol{x}} \in \mathbb{R}^{M \times H \times W}$ denote the corresponding ensemble predictions with size $M$. $H$ and $W$ represent the number of grid points in the latitude and longitude dimensions, respectively. The definitions of the metrics below follow common practices WeatherBench 2 (Rasp et al., 2024).

**Area weighting.** We follow the standard practice of area-weighting all spatially aggregated metrics to account for the convergence of meridians at the poles (Rasp et al., 2024). The unnormalized area weights are computed as $\tilde{a}_h = \sin \phi_h^u - \sin \phi_h^l$, where $\phi_h^u$ and $\phi_h^l$ represent the upper and lower latitude bounds, respectively, for the grid cell with latitude index $h \in \{1, \ldots, H\}$. The normalized area weights $a_h$, used in all metrics below, are defined as: $a_h = \frac{\tilde{a}_h}{\frac{1}{H} \sum_{h'=1}^{H} \tilde{a}_{h'}}$.

We complement our primary evaluation metric, the CRPS introduced in Section 3.2, with the metrics below:

**Ensemble-mean RMSE.** The ensemble-mean RMSE is computed based on the ensemble mean prediction, $\bar{\boldsymbol{x}} = \frac{1}{M} \sum_{m=1}^{M} \hat{\boldsymbol{x}}^{(m)}$. The ensemble-mean RMSE is defined as:

$$\text{RMSE} = \sqrt{\sum_{h=1}^{H} \sum_{w=1}^{W} a_h \left( \bar{\boldsymbol{x}}_{h,w} - \boldsymbol{x}_{h,w} \right)^2}. \tag{6}$$

**Spread-Skill Ratio (SSR).** The spread-skill ratio is defined as the ratio between the ensemble spread and the ensemble-mean RMSE (Fortin et al., 2014), where the spread is calculated as the square root of the spatially averaged ensemble variance. Let $\sigma_{h,w}^2$ denote the variance of the ensemble predictions at grid point $(h, w)$. The Spread and SSR are defined as:

$$\text{Spread} = \sqrt{\sum_{h=1}^{H} \sum_{w=1}^{W} a_h \sigma_{h,w}^2}, \qquad \text{SSR} = \sqrt{\frac{M+1}{M}} \frac{\text{Spread}}{\text{RMSE}}, \tag{7}$$

where the term $\sqrt{(M+1)/M}$ is a correction factor that accounts for the bias in the RMSE of the ensemble mean when using a finite ensemble size $M$. The SSR serves as a measure of ensemble reliability (calibration), where values closer to 1 are better. Values smaller (larger) than 1 indicate underdispersion (overdispersion).

# C. Detailed Results

## C.1. Computational Comparison Details (Figure 1)

The Pareto figures visualize the compute requirements listed in Table 2, complemented by the bulk skill metric defined below. Figure 1 in the main paper plots skill against inference latency, with training compute encoded as bubble size. Figure 6 below complements it in two ways: it additionally shows ensemble-mean RMSE improvement on the right subfigure, and moves training compute to the x-axis, with bubble size now encoding inference speed. We choose to do the latter because inference latency is unclear for some baselines in the RMSE comparison. All reported inference times correspond to a single-member, 60-step rollout.

**Compute normalization.**   To facilitate a fair comparison across heterogeneous hardware, we normalize all reported training and inference times from Table 2 to "Estimated A100-Hours." Following the conversion standards in Couairon et al. (2026), we adopt the following equivalence factors: 1 V100 $\approx$ 0.5 A100, 1 H100 $\approx$ 2 A100, 1 TPUv4 $\approx$ 1.5 A100, and 1 TPUv5e $\approx$ 3 A100. We extend this framework by estimating 1 H200 $\approx$ 2.5 A100. Furthermore, to account for resolution discrepancies, we scale down the training and inference costs of $1°$ models (GenCast) by a factor of 1.5 to more fairly compare against $1.5°$ baselines. We note that this linear scaling is a rough approximation and may not hold for architectures with super-linear complexity, such as NeuralGCM ENS.

**Bulk skill metric definition.**   The y-axes of Figure 1 quantify the mean CRPS and ensemble-mean RMSE improvement relative to the operational IFS ENS baseline. All scores are computed at a spatial resolution of $1.5°$ (regridded from $0.25°$ for GenCast and IFS ENS). Following the evaluation protocol of Couairon et al. (2026), we aggregate performance across five representative upper-air variables: geopotential at 500 hPa (z500), specific humidity at 700 hPa (q700), temperature at 850 hPa (t850), and the u- and v-components of wind at 850 hPa (u850, v850). The final score is derived by averaging the relative improvement across all variables and daily lead times up to 10 days ($\tau \in \{24, 48, \ldots, 240\}$ hours). All evaluations are conducted on the full 2020 test set (732 initializations at 00z and 12z), with baseline scores sourced directly from the WeatherBench2 leaderboard (Rasp et al., 2024) to ensure rigorous comparison. In the ensemble-mean RMSE plot, we include the deterministic GraphCast and FuXi as well as the stochastic Stormer baseline for reference. Because FuXi results for q700 are not available on Weatherbench2, we exclude this variable for the RMSE plot. Its inclusion would only minimally change the figure.

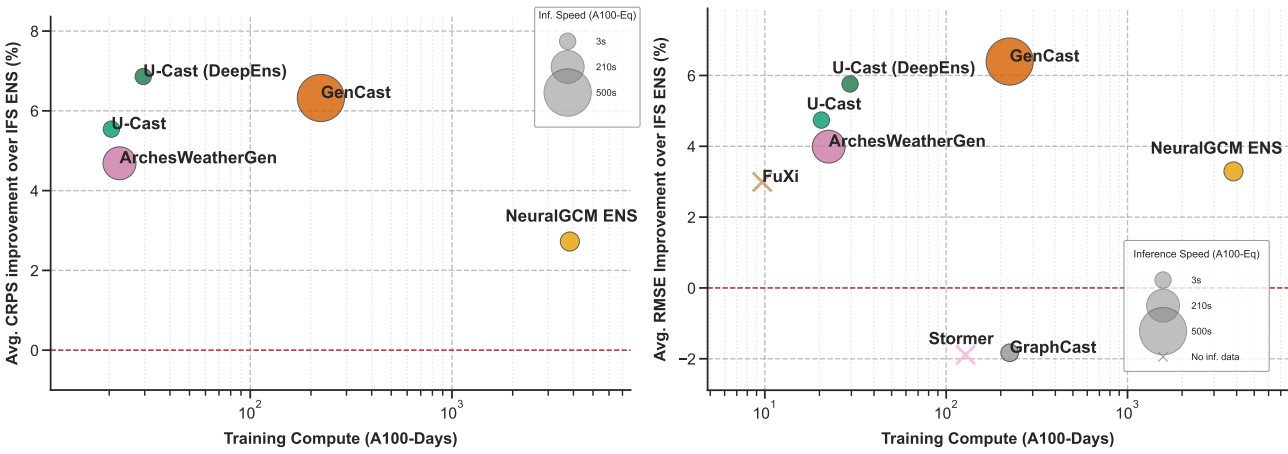

*Figure 6.* **The Efficiency-Accuracy Pareto Frontier.** We visualize forecast skill (y-axis, % improvement over IFS ENS in terms of CRPS on the left and RMSE on the right), training cost (x-axis), and inference latency (bubble size). Our model (top-left) achieves state-of-the-art performance while requiring ***an order of magnitude less compute for training or inference*** compared to leading baselines.

## C.2. Computational Comparison Details (Table 2)

With the exception of the training cost for $1°$ FGN (whose estimate we explain below), all training and 60-step rollout inference compute requirements are directly taken from the respective papers:

- ArchesWeatherGen (Couairon et al., 2026): The best ArchesWeatherGen model is trained in $\approx 45$ V100 days and

"generates 15-day weather trajectories (with 24h time steps) at a rate of 1 minute per ensemble member on an A100 GPU card". Thus, to complete a 60-step forecast, it would need $4\times$ as much or around 4 minutes for inference. The paper also explains that the model "requires 3.5s per 24h forecast on an A100 card". Multiplied by 60, this results in 3.5 minutes for a 60-step forecast, which is the quantity that we use in Table 2 and Section C.1.

- NeuralGCM ENS (Kochkov et al., 2024): Table G6 in the paper reports using 128 TPUv5e's for 10 days to train the ENS version at $1.4°$ resolution, and generating a 10-day forecast in 12.4 seconds. We convert it to the would-be cost for a 15-day forecast: $18.6 = 1.5 \times 12.4$ seconds.

- Swift (Stock et al., 2025): Models were trained for a duration of 3 days on a cluster of 120 Intel Max 1550 GPUs (64GB), totaling approximately 360 GPU-days. Inference latency was not reported.

- Mosaic (Zhdanov et al., 2026): Training was performed on 8 H100 GPUs over two days. The authors report that a 24-member, 10-day forecast completes in 11.6 seconds on a single H100. Since Mosaic operates at 24-hour temporal resolution, this corresponds to a 10-step rollout, which extrapolates to 69.6 seconds for a 60-step rollout. Under the generous assumption that inference cost scales linearly with ensemble size, this yields a per-member cost of 2.9 seconds.

- GenCast (Price et al., 2024): Training followed a multi-stage protocol: initial pre-training at $1°$ resolution (3.5 days on 32 TPUv5 instances; $\approx 112$ TPU-days), followed by fine-tuning at $0.25°$ (1.5 days on 32 TPUv5 instances; $\approx 48$ TPU-days). Regarding inference, the authors report that the $0.25°$ model generates a 15-day forecast in approximately 8 minutes on a single Cloud TPUv5. Since the model operates with a 12-hour time step, a standard 60-step rollout requires doubling this duration to approximately 16 minutes. For the $1°$ variant, we measure inference latency directly on our own GPU (H200) hardware, which is known to be slower than TPUs for this model due to software inefficiencies.

- AIFS-CRPS (Lang et al., 2024): The O96 ($\approx 1°$) version is trained for 4 days on 64 H100 64GB GPUs (256 H100-days). The N320 ($\approx 0.25°$) version is trained for 7 days on 128 H100 64GB GPUs (896 H100-days). The particularly high cost of the O96 is likely due to training with four ensemble members, while N320 only uses two. A 60-step forecast (15-day forecast, since AIFS-CRPS runs at 6-hourly resolution) is completed in 1 and 4 minutes, respectively.

- FourCastNet 3 (Bonev et al., 2025): The model was trained on 1024 H100 GPUs for 78 hours (3,328 H100-days), followed by two fine-tuning stages (15 hours on 512 A100s and 8 hours on 256 H100s). This substantial training budget is primarily driven by the requirement to generate 16-member ensembles at each step to optimize the CRPS objective. In Table 2, we conservatively report only the base pre-training cost. Regarding inference, the authors report a runtime of under 4 minutes for a 60-day global forecast ($0.25°$ resolution, 6-hour steps). Normalizing this to a standard 60-step rollout (equivalent to a 15-day forecast), we estimate the inference latency to be approximately 1 minute.

- IFS ENS (ECMWF, 2019) is the leading operational numerical weather prediction model. Thus, there is no training cost associated with it. The inference cost is high, however, requiring 1 hour of runtime on 96 AMD CPUs[1].

- FGN (Alet et al., 2025): The authors report that training each of the four ensemble members requires approximately 3 wall-clock days, consuming a combined total of 490 TPUv5p and TPUv6e chip-days per model. In the absence of a specific hardware breakdown, we conservatively standardize this cost to TPUv5p-days in Table 2. Regarding inference, the model generates a 15-day forecast (60 steps at 6h resolution) in just under 1 minute on a single TPUv5p. As the model is proprietary, specific inference latencies for the $1°$ ablation are unavailable. Its training cost is estimated next.

**Estimate of $1°$ FGN training cost.** FGN (Alet et al., 2025) and GenCast (Price et al., 2024) share a similar graph transformer architecture and training curriculum: pre-training a $1°$ resolution model followed by fine-tuning at $0.25°$. While the FGN paper reports a total cost of 490 TPU days, they do not provide a breakdown by resolution. Therefore, we reference GenCast, which required 112 TPU days for $1°$ and 48 TPU days for $0.25°$ resolution, to estimate the allocation. In GenCast, approximately 70% of compute was dedicated to $1°$ pre-training. Applying this ratio to FGN yields 343 TPU days. However, FGN also includes a final autoregressive fine-tuning stage at $0.25°$. Because this stage involves backpropagating gradients through rollouts—a computationally expensive process—we conservatively adjust the estimate for $1°$ pre-training down to 300 TPU days. Note that this estimate is for *one* out of the 4 deep ensemble models that it uses.

---

[1]See https://www.ecmwf.int/en/newsletter/181/earth-system-science/data-driven-ensemble-forecasting-aifs

**U-Cast training cost breakdown.**  At $1.5°$, our model requires 7 H200-days for the deterministic pre-training phase, followed by 1.2 H200-days for the probabilistic CRPS fine-tuning phase. For U-Cast's deep ensemble version, the same deterministic pre-training run is used; only the fine-tuning phase is repeated with different random seeds. Thus, the training cost for the 4-member deep ensemble version of (U-Cast DE) is 11.8 H200-days. We highlight that after deterministic pre-training is complete, training the probabilistic model only takes a fraction of the time. This also enables the efficient training of a deep ensemble, in contrast to models like FGN, where each deep ensemble member requires repeating the *full* training procedure. For informative purposes, we also trained U-Cast at $1°$ resolution. At this higher-resolution, the training cost increases to 13 H200-days for deterministic pre-training and 1.89 H200-days for CRPS-based fine-tuning.

**U-Cast inference latency.**  We benchmarked the time needed for U-Cast to generate a single 60-step, i.e., 30-day, forecast on both an H100 and an A100 GPU. The timing per rollout includes the time needed to forward the inputs through the network, denormalize the predictions, update the inputs for the next autoregressive step, and accumulate the predictions (and moving them to the CPU) to return them. Batching ensemble members scales sublinearly. For example, U-Cast completes a 60-step rollout on an H100 in 2 seconds for a single member versus 12 seconds for ten members. For efficiency, the forward pass is computed in mixed precision, and the model architecture is compiled with `torch.compile`. The timings are averaged out over six rollouts, discarding the first one to account for the one-time compilation and spin-up overheads. After the slower spin-up rollout, compilation speeds up inference latency by close to $50\%$.

## C.3. A Note on Resolution Choice

Our primary evaluation uses $1.5°$ resolution, the standard WeatherBench 2 benchmark grid. We additionally trained a $1°$ model, whose computational costs are reported alongside the $1.5°$ model in Section C.2. A direct $1°$ comparison to GenCast's open-source $1°$ checkpoint was infeasible because GenCast's $1°$ model was trained on subsampling-regridded data rather than WeatherBench 2's first-order conservative regridding, precluding fair evaluation at either $1°$ or $1.5°$.

## C.4. Extended Metrics: Absolute CRPS, RMSE, and Spread-skill Ratio

The WeatherBench 2 benchmark limits comparisons to surface variables and the 500, 700, and 850 hPa pressure levels. Accordingly, our analysis in this section is restricted to this subset of variables. All metrics are computed based on 50-member ensembles.

We present comprehensive quantitative results on the $1.5°$ WeatherBench 2 benchmark in Figure 7 (Absolute CRPS), Figure 8 (Ensemble-Mean RMSE relative to IFS ENS), and Figure 9 (Spread-Skill Ratio; SSR), evaluated on all 732 initial conditions from 2020. The relative RMSE results largely corroborate the findings from our main CRPS analysis in Section 4.2, although the GenCast baseline does perform relatively stronger in terms of ensemble-mean RMSE than in terms of CRPS. Regarding calibration, the SSR indicates that U-Cast tends to be under-dispersive (overconfident), particularly at short-to-medium forecast horizons for geopotential, mean sea level pressure, and surface winds. However, for variables such as 2-meter temperature and specific humidity (700 and 500 hPa), our ensemble dispersion is comparable to that of the operational IFS ENS. We note that among the evaluated baselines, ArchesWeatherGen, followed by GenCast, tend to exhibit the most well-dispersed ensembles.

## C.5. Evaluation on year 2022

To assess a basic kind of generalization, we repeat the evaluation from Section C.4 on the test year 2022 (Figure 10), aggregating results over all 730 initial conditions (all `00z` and `12z` times). This represents a challenging test, as our model was trained only on data up to the end of 2019. Despite this, U-Cast largely reproduces the strong performance seen in the 2020 evaluation, with only small degradation in specific variables (e.g., `z500`) that could likely be addressed by fine-tuning U-Cast all the way up to the end of 2021.

## C.6. U-Cast vs U-Cast DeepEns

In Figure 11, we isolate the effect of deep ensembling by comparing the single-seed U-Cast model against its $K = 4$ deep ensemble variant (U-Cast DE), both evaluated on the 2020 test set. Deep ensembling yields consistent CRPS improvements across nearly all variables and lead times, with the largest gains concentrated in short-to-mid-range geopotential (up to 4% for `z500` at 1–5 day lead times) and stratospheric variables. The improvement is most pronounced for variables where the single model already performs well relative to baselines, suggesting that deep ensembling primarily reduces residual

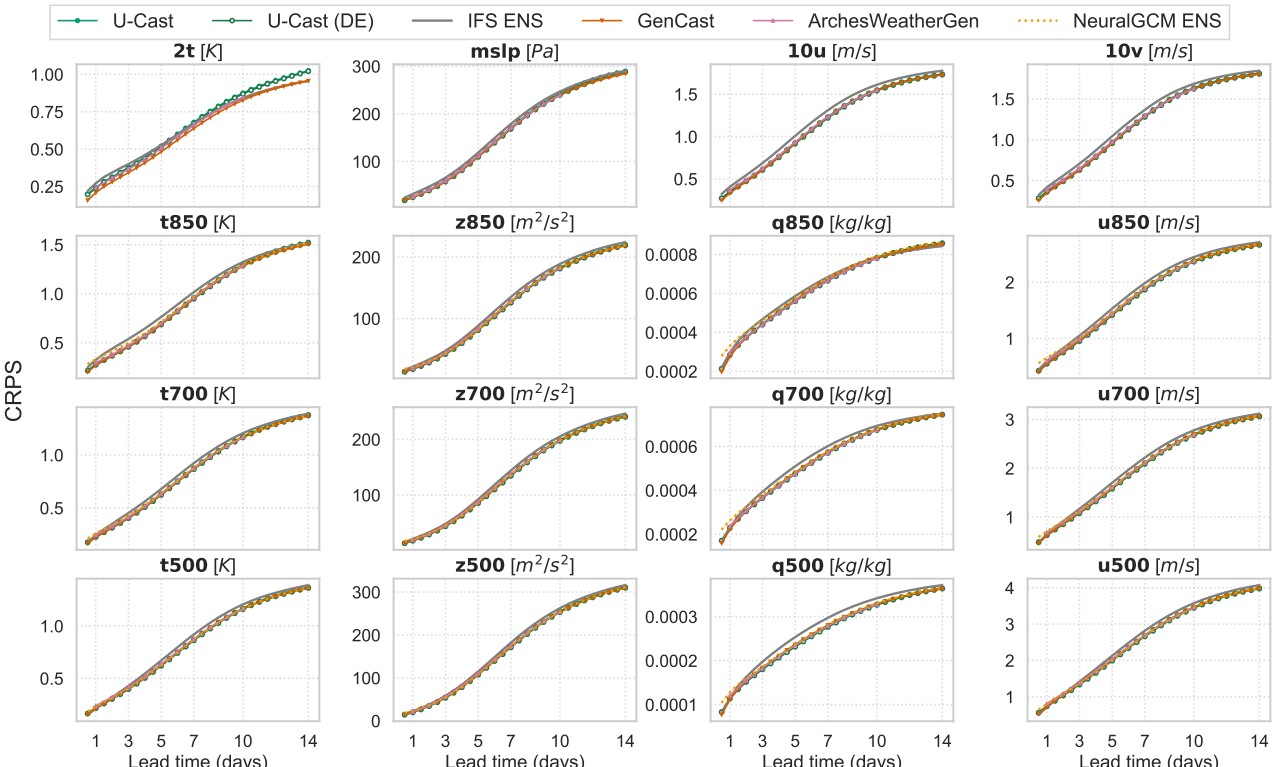

*Figure 7.* **WeatherBench 2 Comparison** ($1.5°$ **resolution**): **Absolute CRPS.** We report the CRPS skill as a function of forecast horizon (lower is better). This figure visualizes the same as Figure 3 but in absolute terms, without normalizing by the performance of IFS ENS.

sampling noise and improves tail calibration rather than correcting systematic biases.

Importantly, the marginal training cost of deep ensembling in U-Cast is minimal: each additional fine-tuned member requires only 1.2 H200-days (the cost of repeating Stage 2), since all members share the same deterministic backbone from Stage 1. The full 4-member deep ensemble thus costs only 11.8 H200-days in total, compared to the $4 \times 490 = 1{,}960$ TPU-days that would be required for an equivalent FGN deep ensemble (Alet et al., 2025) (at $0.25°$ resolution). This efficiency makes deep ensembling a practical default rather than a luxury reserved for well-resourced labs.

## C.7. Power Spectra

In Figure 13, we compare the spectral density of 10-day U-Cast forecasts against ERA5 reanalysis, averaged over mid-latitudes ($[25°, 55°]$). U-Cast reproduces realistic spectra for surface and specific humidity variables, but tends to produce excess power at high spatial frequencies for geopotential, temperature, and wind fields. This suggests that while the model captures large-scale atmospheric structure well, it occasionally generates spurious small-scale variability. Such spectral artefacts have been connected to residual prediction training (Bonev et al., 2025; Zhdanov et al., 2026), so training U-Cast on a full-state prediction objective would be a natural next step to address this limitation of U-Cast.

## C.8. Qualitative Forecast Analysis

Figure 14 and Figure 15 present representative forecast trajectories generated by U-Cast, initialized on 2020-07-14 `00z`. Qualitatively, the model maintains realistic atmospheric structure and sharp gradients even at extended lead times (e.g., 14 days). As might be expected from the imperfect power spectra identified in Section C.7, however, an analysis of the error fields reveals systematic artifacts. In particular, we note such artifacts in the polar regions, most notably visible as distinct bias patterns in the Antarctic 2-meter temperature (Figure 15b). The emergence of these anomalies at short forecast horizons (day 3) indicates that they likely stem from fundamental architectural or training factors—such as the treatment of polar boundary conditions—rather than autoregressive drift or the absence of fine-tuning. Addressing these topological edge cases remains a targeted area for future architectural refinement.

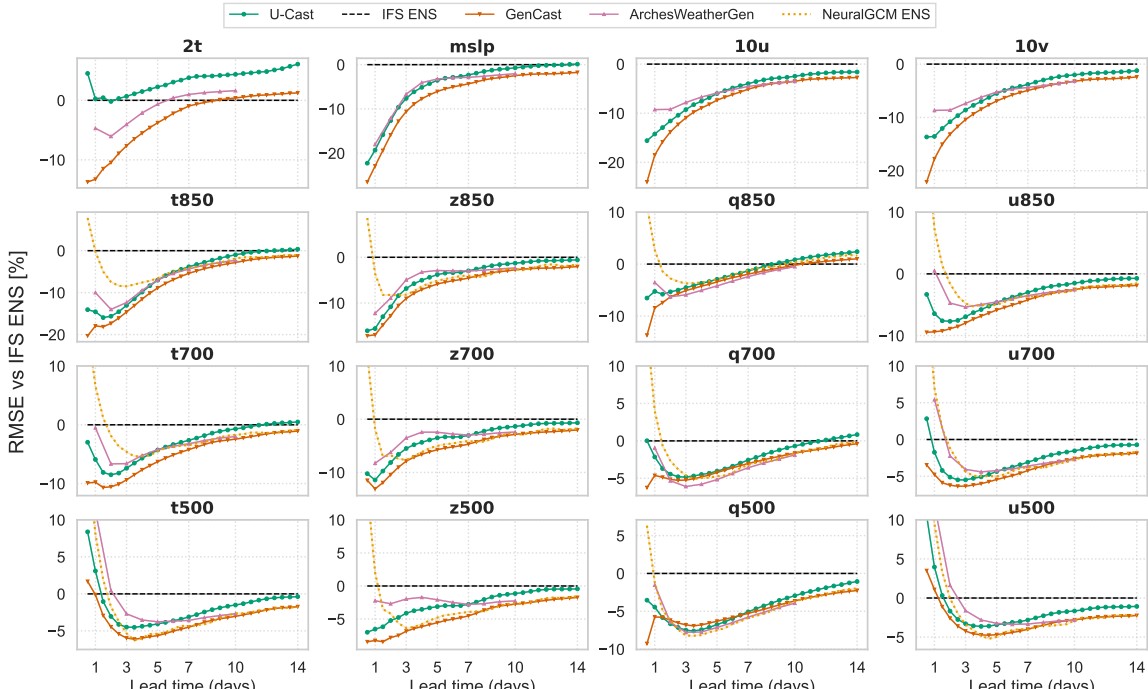

*Figure 8.* **WeatherBench 2 Comparison** ($1.5°$ **resolution**): **RMSE.** We report the 50-member ensemble-mean RMSE skill relative to the operational IFS ENS (%, lower is better) as a function of forecast horizon. Baseline scores are sourced directly from the official leaderboard (Rasp et al., 2024). Numbers after the variable abbreviations refer to the pressure level in hPa. Note that the GenCast baseline uses native $0.25°$ forecasts regridded to $1.5°$, which may confer an advantage over models strictly operating at $1.5°$ (like U-Cast, ArchesWeatherGen, and NeuralGCM). Relative to the CRPS results, GenCast performs stronger in terms of RMSE than U-Cast.

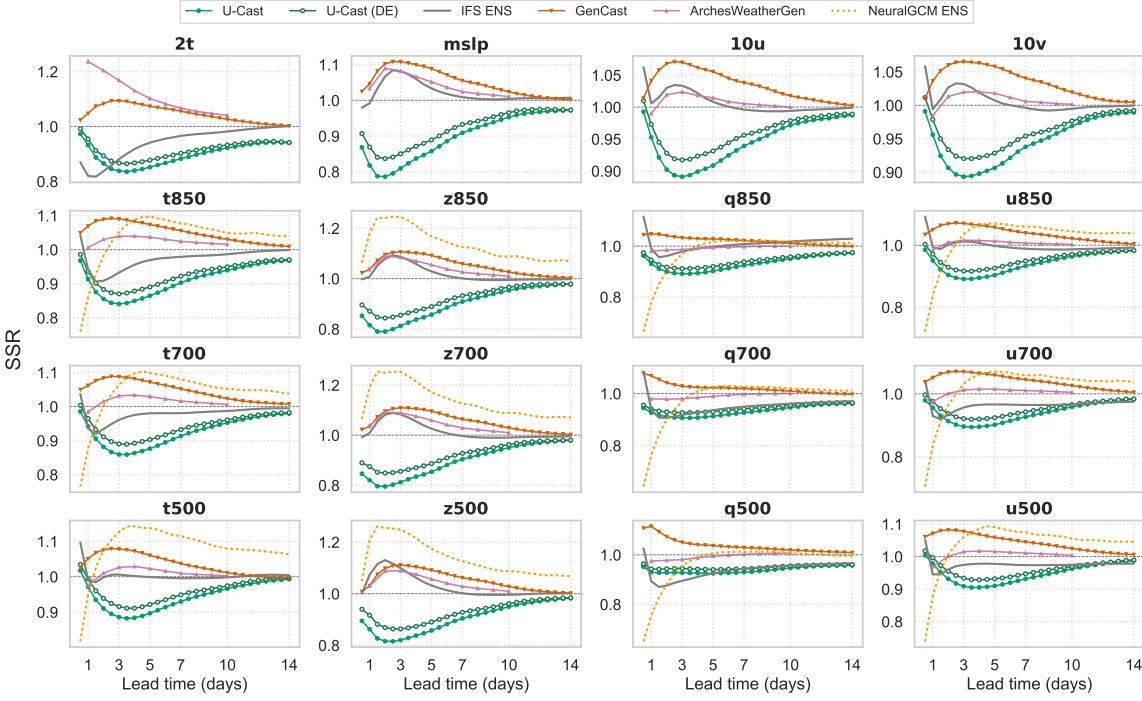

*Figure 9.* **WeatherBench 2 Comparison** ($1.5°$ **resolution**): **SSR.** We report the Spread-Skill ratio skill as a function of forecast horizon (closer to 1 is better). Baseline scores are sourced directly from the official leaderboard (Rasp et al., 2024). Numbers after the variable abbreviations refer to the pressure level in hPa. U-Cast generates more overconfident forecasts than the baselines, especially in the 1-to-7-day regime. U-Cast (DE) improves calibration, with SSR rarely falling below 0.85 and achieving better short-range calibration than NeuralGCM ENS for several variables.

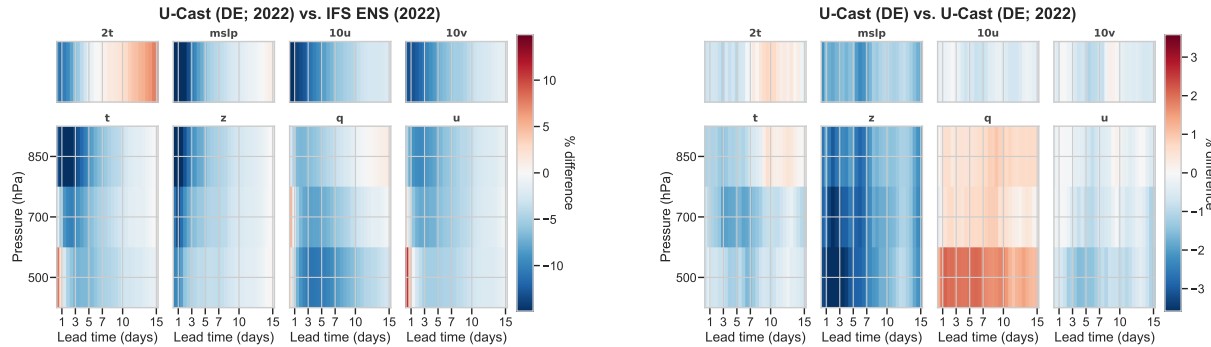

*Figure 10.* **Comparison of U-Cast (DE) evaluated on 2022 against IFS ENS (left) and against U-Cast (DE) evaluated on 2020 (right).** Blue indicates that U-Cast achieves a lower (better) CRPS, while red favors the baseline. U-Cast consistently outperforms IFS ENS on 91.5% of metrics, with notable exceptions in long-range 2-meter temperature and a few variables at the 12-hour lead time (e.g., a 10.8% deficit in u500). On average, U-Cast improves upon IFS ENS by 4.45%, with the strongest gains ($\approx$ 18–22%) observed at 12-to-36-hour lead times for mslp. These results are comparable to those obtained for the test year 2020 (Figure 2), demonstrating that U-Cast, despite being trained only on 1979–2019 data, generalizes well to more distant years. This robustness is further supported by the right panel, where the maximum degradation of U-Cast (DE; 2022) relative to its 2020 performance remains below 4.5%, concentrated in geopotential variables (especially z500), while specific humidity variables even show improved CRPS scores (up to 2%).

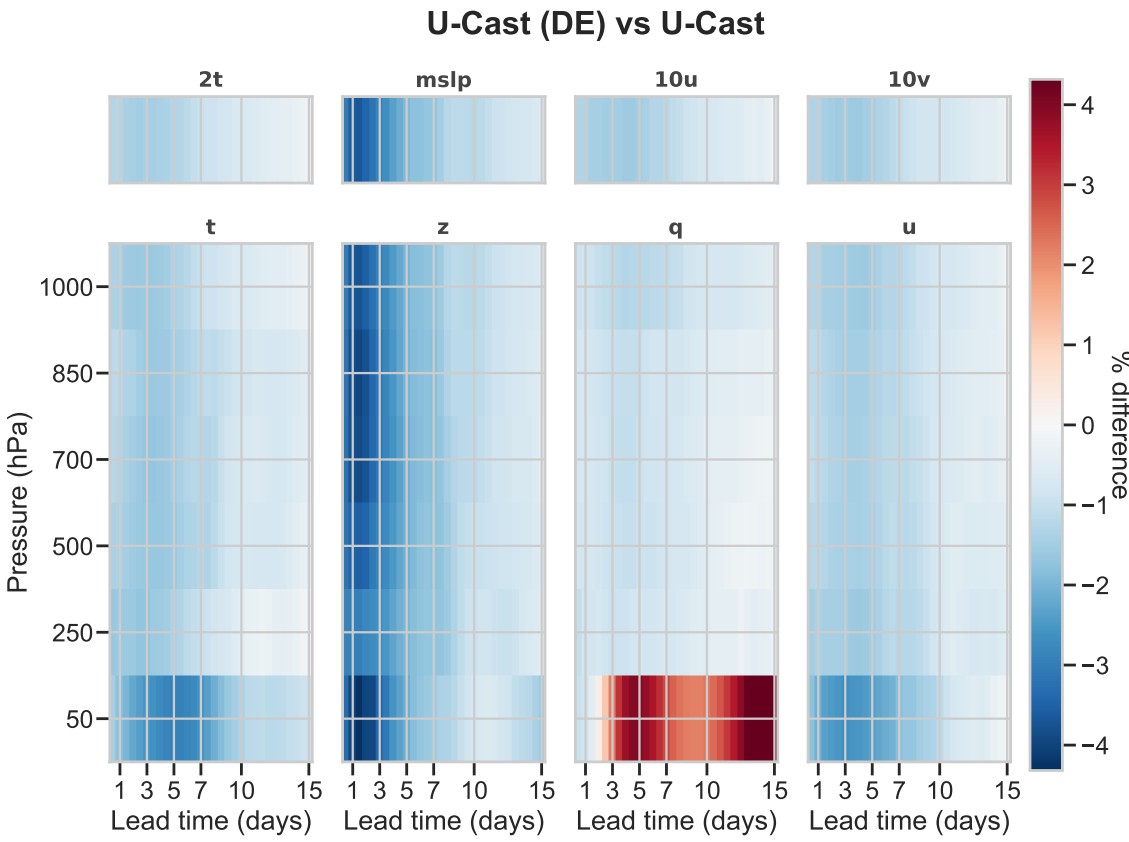

*Figure 11.* **Score card comparison of U-Cast (DeepEns) vs. U-Cast.** Deep ensembling U-Cast via fine-tuning four different versions of it consistently improves CRPS scores, especially for short-to-mid-range geopotential and stratospheric (by up to 4%; except q50) variables.

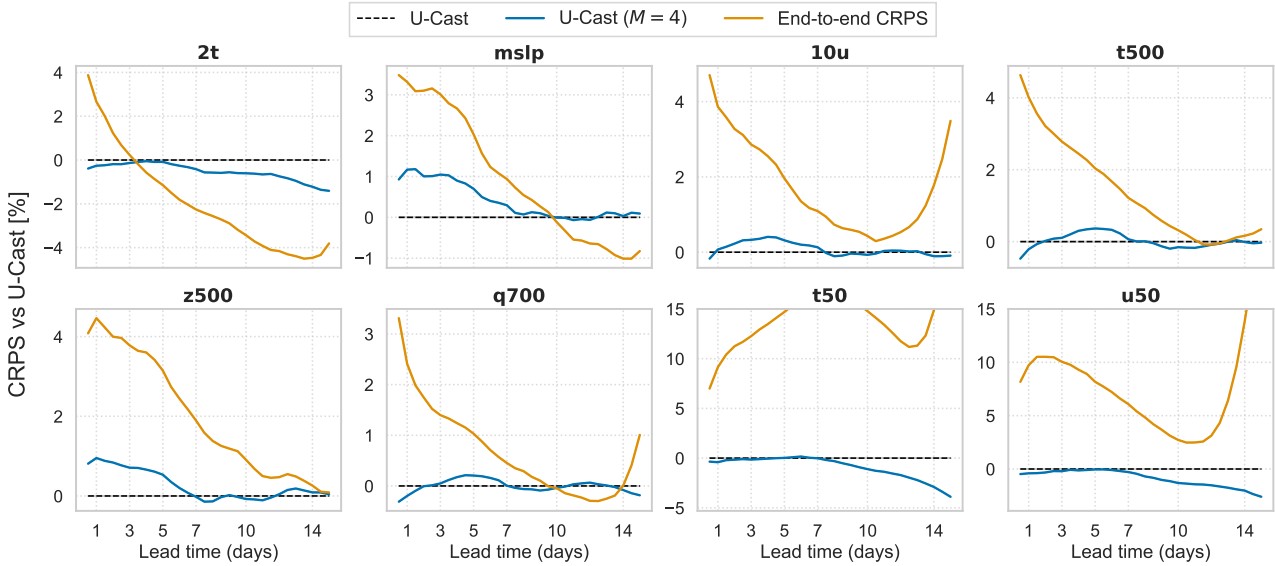

*Figure 12.* **Curriculum and ensemble-size ablations (full evaluation).** Relative CRPS vs. U-Cast across variables and lead times (higher means worse than U-Cast). *End-to-end CRPS* (orange) trains from scratch without deterministic pre-training; it consistently degrades short-range CRPS by 3–5% and stratospheric variables by 5–15% across all lead times, while recovering or slightly improving long-range scores for select variables (e.g., 2t). *U-Cast (M = 4)* (blue) doubles the training ensemble size; it yields only marginal improvements despite doubling per-step cost, with modest gains concentrated at long-range stratospheric variables (up to 3% for t50).

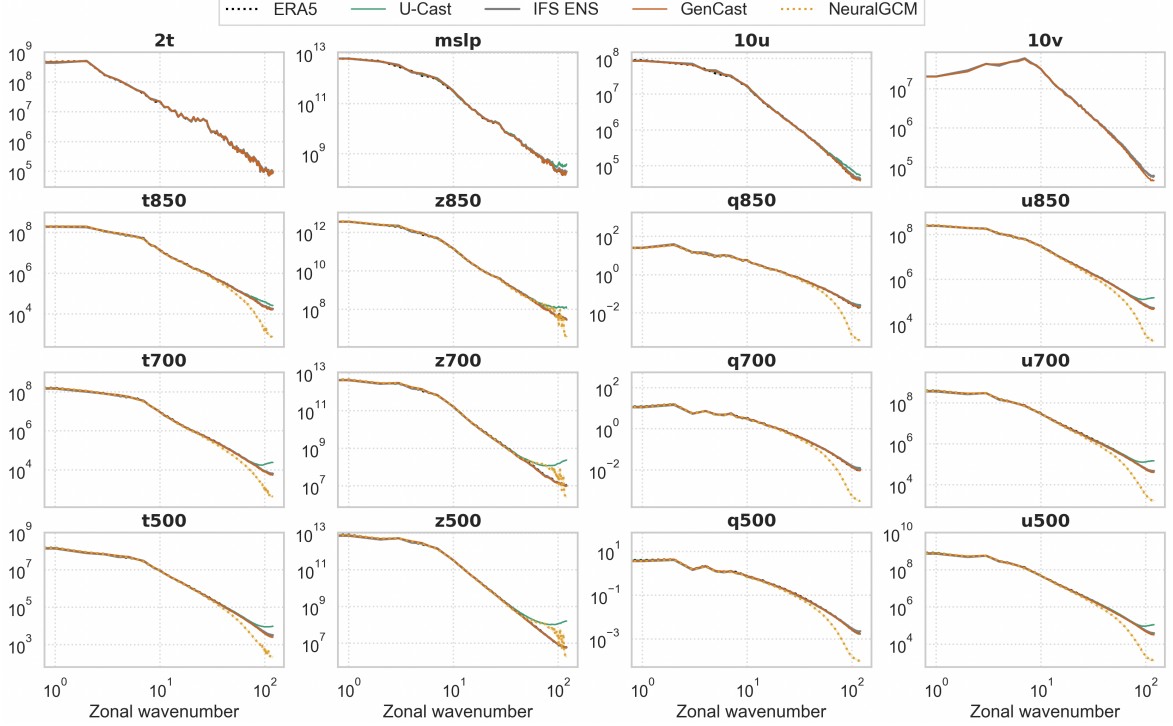

*Figure 13.* Spectral density of 10-day forecasts, averaged over mid latitudes ($[25°, 55°]$). While U-Cast generates realistic spectra for the surface and specific humidity variables, it tends to generate excess power at high frequencies for the other variables.

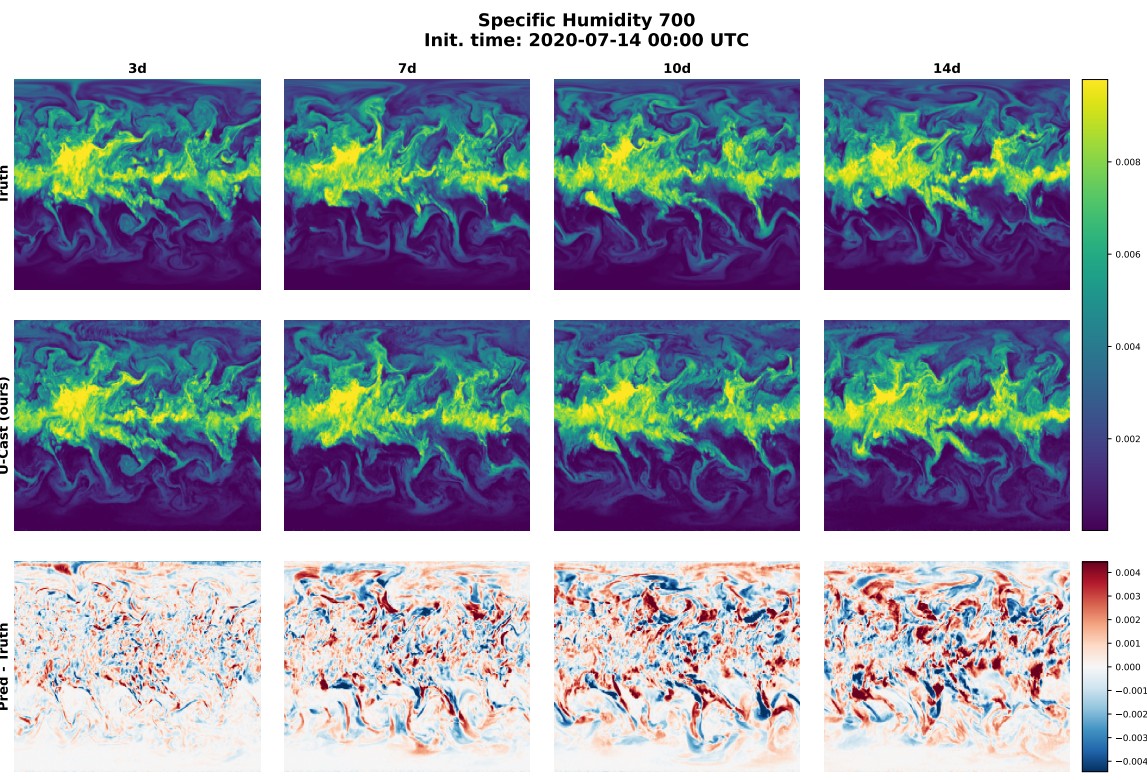

*Figure 14.* Example visualizations of U-Cast (second row), the corresponding ground truth (first row), and the bias (last row) for specific humidity at 700 hPa (q700) and forecast lead times 3, 7, 10, and 14 days.

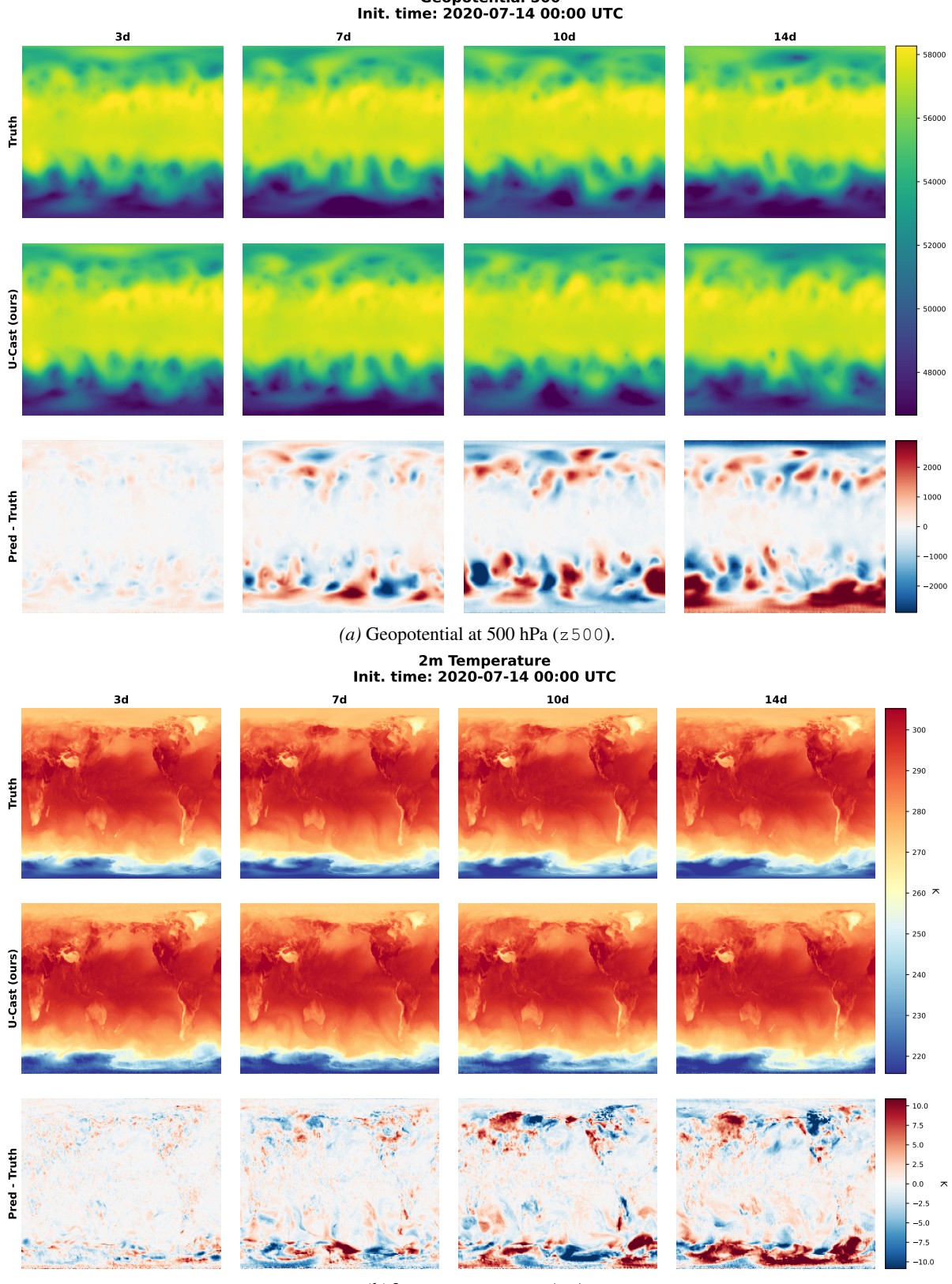

*(a)* Geopotential at 500 hPa (z500).

*(b)* 2-meter temperature (2t).

*Figure 15.* Example visualizations of U-Cast (second row), the corresponding ground truth (first row), and the bias (last row) for two example variables and forecast lead times 3, 7, 10, and 14 days.

