# OpenReview forum: "U-Cast: A Surprisingly Simple and Efficient Frontier Probabilistic AI Weather Forecaster"
_ICML.cc/2026/Conference — ICML 2026 regular_

### Official Review · Reviewer_Q4Cq · 2026-02-15

**Soundness:** 3
**Presentation:** 3
**Significance:** 3
**Originality:** 3
**Overall Recommendation:** 4
**Confidence:** 5

**Summary:**

This paper presents U-Cast, a streamlined probabilistic forecaster based on the standard UNet architecture. U-Cast leverages Monte Carlo Dropout for probabilistic predictions and is trained using the Muon optimizer. With a two-stage curriculum training strategy, U-Cast achieves SOTA performance at a lower computational cost, showing that general-purpose architectures can outperform specialized designs.

**Compliance With Llm Reviewing Policy:**

Affirmed.

**Final Justification:**

Thank the authors for their response. It addressed my concerns, and I've updated my score accordingly.

**Key Questions For Authors:**

* U-Cast employs the Monte Carlo method for ensemble forecasting, which seems almost identical to the ensemble forecasting approach used by FuXi. However, the forecasting performance differs significantly. Could some analytical experiments be included to demonstrate which specific aspect plays a role, rather than merely providing a textual explanation?
* To demonstrate that the complexity of the meteorological forecast model is not necessary, it should be proven as much as possible in the same environment that simple models can outperform complex ones. However, the U-Cast model proposed in the article uses the Muon optimizer, while most baseline models are based on the AdamW optimizer. Why not use the same optimizer to eliminate the influence of factors other than architecture? The ablation experiment in the article shows that using the AdamW optimizer leads to a significant performance drop. Then, can complex models also benefit from the Muon optimizer and still maintain an advantage over simple models in terms of performance?

**Limitations:**

Yes

**Strengths And Weaknesses:**

Strengths:
This article presents a probabilistic weather forecast method that is simple enough and highly efficient, outperforming existing complex weather forecast models in terms of performance, while avoiding the high computational cost brought about by generative ensemble forecasting.

Weaknesses:
* Too few baselines are included, models like FGN should be added.
* Curriculum learning strategies lack analysis and ablation, supplementary experiments should be added.
* The major performance gain appears to stem from the use of the Muon optimizer. It remains unclear, however, whether a U-Net trained with Adam/AdamW still outperforms baseline models. This comparison should be explicitly reported.

---

> ### Author Rebuttal · Authors · 2026-03-31
>
> We thank Reviewer Q4Cq for the detailed review and for noting our method's "high efficiency" and that it outperforms "existing complex weather forecast models." We address each concern below.
>
> **W1: Baselines / FGN.** We respectfully note that in Section 4.3 we include *all* baselines provided by the standard WeatherBench 2 benchmark probabilistic leaderboard. FGN is a closed-source model (no public code or weights), so direct evaluation is not possible. However, our ablation in Fig. 4 applies one of FGN's core components—learned perturbations via adaLN noise injection—within our framework and finds that it deteriorates CRPS by up to 3–4% while improving calibration (SSR). This suggests our simpler MC Dropout approach offers a favorable skill–calibration trade-off.
>
> **W2: Curriculum ablation.** We acknowledge this should have been ablated in the submission. **[New]** To address this, we retrained U-Cast from scratch on CRPS only, using the same total training budget (in H200-days; allowing us to train it for 50 epochs on the CRPS from scratch).
> The curriculum tends to generally yield 3-4% better CRPS, and in the case of stratospheric variables like t50, up to 5-15% better (see figure at this anonymous URL: https://drive.google.com/file/d/1ZAWwXUYSLWyVq-Pvev5jJG_MQsJ3AX1q/view?usp=sharing). This shows it is a meaningful contribution that we recommend in practice. The from-scratch model does recover competitively at long lead times for select variables, but the short-range and stratospheric degradation is consistent and substantial. We will include this important ablation in the main body of our revised paper.
>
> **W3/Q2: Muon optimizer.**
> We thank the reviewer for this insightful point and will revise the manuscript to more clearly frame the contribution as a recipe rather than attributing results to architecture alone. For example, the abstract now concludes with "scalable, general-purpose architectures *paired with efficient training curricula* can match complex domain-specific designs at a fraction of the cost", replacing the original version's imprecise framing. We will further strengthen this throughout, including in the discussion and conclusion.
>
> To clarify the empirical picture: Muon is not clearly superior to AdamW for the deterministic pre-training stage given sufficient training time. The large performance drop in Figure 4 arises because that ablation replaces Muon with AdamW for *both* stages, and the deficit is concentrated in the probabilistic fine-tuning stage, where Muon enables convergence within our curriculum's brief 8-epoch budget. We will add this clarification to the revision.
>
> More fundamentally, our contribution is not that any single component is solely responsible for U-Cast's performance, but the *synergy of the full recipe*: The curriculum keeps the expensive CRPS phase to 15% of total compute, and Muon ensures this short phase actually converges to frontier quality. We expect that a well-tuned AdamW pipeline trained end-to-end on CRPS from scratch, without the curriculum, could eventually reach comparable skill given enough compute. However, this would negate the extreme training efficiency that is central to our contribution, and verifying this hypothesis lies outside our compute budget.
>
> **Q1: FuXi is not similar to U-Cast.** We believe there may be a misunderstanding: FuXi-ENS (Zhong et al., Science Advances 2025) is built quite differently from U-Cast. We use MC Dropout + CRPS loss for stochasticity (model-based), while FuXi-ENS uses VAE-based input perturbations with a deterministic forecast model (input-based). These are orthogonal approaches that are, in fact, complementary. Our results suggest that model-based stochasticity provides more value per unit complexity (e.g., our SSR never drops below 0.8 for z500, while FuXi-ENS drops to ~0.7). Nothing prevents future work from combining both approaches.

---

> > ### Author Rebuttal · Reviewer_Q4Cq · 2026-04-04
> >
> > Figure 4 demonstrates the substantial performance superiority of the Muon optimizer over AdamW (a 15% improvement on Z500). However, the argument for "accelerated convergence" would be significantly more convincing if the authors could include a comparison of the performance curves between AdamW and Muon under sufficient training steps.
> > Regarding Q1, I would like to clarify that there appears to be a misunderstanding. The "FuXi" mentioned refers specifically to "FuXi: A cascade machine learning forecasting system for 15-day global weather forecast", rather than FuXi-ENS. Please refer to Section 3.4 of the original manuscript, where explicitly detail how FuXi utilizes MC Dropout for ensemble forecasting.

---

> > > ### Author Response · Authors · 2026-04-07
> > >
> > > **Figure 4:** This is an excellent suggestion, and we'll add this suggested plot showing the training-time convergence speed of AdamW vs Muon to our revised Appendix. The key takeaway is that the AdamW run requires 4x more training steps to reach the same performance (e.g., z500 CRPS at t=12) early during training, which reduces to only 1.5x towards the end of our trainings.
> > >
> > > **Q1**: Thanks for the clarification about Q1. We had overseen that the original FuXi paper includes an ensemble variant already (besides the newer ensemble version we discussed above). This FuXi paper is actually another perfect example for our claim in Section 3.3. that *"while early works applied Dropout to weather forecasting (Scher & Messori, 2021; Garg et al., 2022; Hu et al., 2023), they reported unsatisfying, underdispersive ensembles"*, as FuXi's strong underdispersion (at longer lead times) is clearly visible in Fig. 5 of the FuXi paper. We'll add FuXi as the 4th example in this citation list.
> > >
> > > As for why FuXi and these other MC Dropout-based prior works are so underdispersed, we note in the same section 3.3. of our paper that *"We hypothesize that this failure was not intrinsic to Dropout, but rather a consequence of training objectives (like MSE) that do not explicitly reward probabilistic skill"*. In particular, FuXi is trained on MAE, while U-Cast is trained on the CRPS loss.

---

### Official Review · Reviewer_boNT · 2026-03-11

**Soundness:** 3
**Presentation:** 4
**Significance:** 3
**Originality:** 3
**Overall Recommendation:** 5
**Confidence:** 3

**Summary:**

Current SoTAs in weather forecasting rely on complex network architectures and computationally expensive training; the authors argue for a simpler approach. The proposed method uses a modified version of U-Net, where the convolution part is used for physical transformation and a newly added self-attention part for non-local interactions. To encourage the model to account for uncertainty efficiently, a two-stage training is designed. Fundamental atmospheric dynamics is learnt through the first stage deterministic pre-training. Then, in the second stage, MC-dropout is used for efficient CRPS training. Muon and AdamW is both used for the optimisation. Experiment results have shown that the proposed method U-Cast outperforms or performs on par with current SoTA with less computation required.

**Compliance With Llm Reviewing Policy:**

Affirmed.

**Final Justification:**

The rebuttal has addressed my concerns regarding the training recipe, rolling sphere into a 2D grid, and the choice of $M$. Therefore I have updated my confidence from 2 to 3.

**Key Questions For Authors:**

1. Why is the method based on the U-Net, not other neural network architectures?

2. If I understand things correctly, only two sampled networks are used for the CRPS stage. I wonder whether increasing $M$ will improve the performance? (Given the limited time, I'm not asking for additional experiments. Just wondering if this is something the authors have tried before),

**Limitations:**

See weakness above.

**Strengths And Weaknesses:**

Strengths:

- The proposed method is simple, well-motivated, and efficient.

- The proposed method performs better or on par with SoTA while being much more computationally efficient.

- The paper is well-written and easy to follow.

Weakness:

- No autoregressive training is used during training, so the long-term prediction deteriorates.

- Rolling the sphere into a 2D grid might be too simple an approximation.

---

> ### Author Rebuttal · Authors · 2026-03-31
>
> We thank Reviewer boNT for the positive assessment, noting our method is "simple, well-motivated, and efficient" with results "better or on par with SoTA." We address the questions and weaknesses below.
>
>
> **W1: No autoregressive training.** Correct, we train on single-step predictions only. Autoregressive fine-tuning is a natural extension that would likely improve long-range stability, as we note in our Limitations. Even without it, U-Cast matches GenCast out to 15 days for most variables.
>
>
> **W2: Spherical approximation.** As noted in our Limitations, a HEALPix U-Net (Karlbauer et al., 2024) would address the spherical approximation mismatch while preserving our framework's simplicity and efficiency. We note that most SOTA methods (GenCast, FGN, FourCastNet 3) also operate on 2D grids rather than modeling the 3D nature of the atmosphere. Follow-up work that efficiently addresses the vertical dimension is an excellent direction.
>
>
>
>
> **Q1: Why U-Net?** The U-Net is uniquely efficient and, we believe, an excellent fit for atmospheric dynamics: weather physics are intrinsically local at short time scales, making the locality and translational equivariance of convolutions a natural match. The hierarchical encoder-decoder with skip connections captures multi-scale interactions, and the architecture's maturity enables rapid iteration on other design choices (curriculum, stochasticity, optimizer). Our results suggest the architecture choice matters less than the training recipe—a finding we believe is itself valuable for the community.
>
>
> **Q2: Increasing $M$.** To clarify: two sampled *predictions* (from the same network, but different dropout masks) are used during CRPS fine-tuning. **[New]** Increasing M is a thoughtful suggestion we explored during the rebuttal. We found $M=4$ to provide limited improvement over $M=2$ (within ±1% for most variables) while doubling fine-tuning time, though it did improve long-range CRPS for some variables, notably 2-meter temperature, suggesting an alternative path to stabilizing long-term predictions besides autoregressive fine-tuning. We will add this ablation to our paper.

---

> > ### Author Rebuttal · Reviewer_boNT · 2026-04-02
> >
> > Thank the authors for their response. It addressed my concerns, and I've updated my confidence score accordingly.

---

### Official Review · Reviewer_SffL · 2026-03-14

**Soundness:** 3
**Presentation:** 3
**Significance:** 3
**Originality:** 2
**Overall Recommendation:** 4
**Confidence:** 3

**Summary:**

The paper introduces a probabilistic weather forecasting model built on a standard U-Net backbone with MC Dropout for stochasticity. The paper combines a few known recipes:
- a two-stage training curriculum: deterministic pre-training with MAE, followed by short probabilistic fine-tuning on CRPS
- the Muon optimizer

The method achieves benchmark scores competitive with GenCast on most variables while requiring only ~15 H200-days of training and 12-second inference for a 60-step forecast. The authors frame this as evidence that complex geometric architectures and diffusion-based stochasticity are not prerequisites for frontier probabilistic weather prediction.

**Compliance With Llm Reviewing Policy:**

Affirmed.

**Final Justification:**

The rebuttal addressed my concerns, particularly the deep ensemble calibration results and the curriculum ablation. Remaining limitations (polar artifacts, no precipitation, 0.25 deg scaling) are acknowledged honestly and don't undermine the central message. I maintain my positive score.

**Key Questions For Authors:**

1. The SSR analysis shows clear overconfidence at 1-3 day lead times. A well-calibrated model at these operationally critical horizons would substantially strengthen the paper. Could this be an artifact of the dropout rate, and if so has this been ablated?

2. You exclude total precipitation from the evaluation. Can you provide even preliminary CRPS results on precipitation? If U-Cast performs poorly on precipitation (a highly non-Gaussian, spatially intermittent variable), this would importantly qualify the generality of the simplicity claim.

**Limitations:**

Yes. The authors discuss polar artifacts, under-dispersion, 2t degradation at long lead times, the absence of autoregressive fine-tuning, and out-of-distribution caveats clearly and honestly. The impact statement appropriately highlights democratization benefits and data-driven limitations.

**Strengths And Weaknesses:**

- *Significant empirical finding.* The general result that a combination of simple techniques can beat domain-specialized models is very valuable and could be a course-correction for the current community trajectory. The 10-100x compute gap between larger models is not trivial, and even if it's optimistic it is directionally meaningful.

- *MC Dropout result overturns prior negative findings.* Earlier work concluded Dropout produces underdispersive weather ensembles. The paper argues convincingly that this was an artifact of MSE training, not Dropout itself, and demonstrates that CRPS training resolves the issue.

- *Thorough evaluation.* Comparisons span GenCast (1 deg), WeatherBench 2 (1.5 deg), FGN, FourCastNet3, and IFS ENS, with ablations over optimizer, stochasticity source, and dropout rate. The inclusion of SSR analysis and qualitative polar artifact diagnosis is commendably honest.

**Weaknesses:**

- *Calibration is a notable deficiency.* The SSR plots show systematic overconfidence at short forecast horizons across most variables. For a probabilistic model, well-calibrated uncertainty is arguably as important as CRPS skill. This weakness is acknowledged but not addressed.

- *Polar artifacts partially undermine the thesis.* The authors argue geometric inductive biases are unnecessary, yet their model produces systematic Antarctic 2t biases that emerge at short lead times. This suggests that treating the Earth as a flat cylinder with circular padding has costs, weakening (but not invalidating) the central claim.

- *No precipitation.* Total precipitation is excluded entirely, which feels like a big omission for a paper claiming frontier status.

- *Limited methodological novelty.* Each component (U-Net, MC Dropout, MAE pretraining, Muon) is well-established. The curriculum is the most novel element, but the idea of deterministic pre-training followed by probabilistic fine-tuning has precedents in other domains. The contribution is primarily in the combination and the resulting empirical surprise.

---

> ### Author Rebuttal · Authors · 2026-03-31
>
> We thank Reviewer SffL for the thoughtful review and for recognizing that our work represents a "significant empirical finding" that "could be a course-correction for the current community trajectory"—that is exactly our goal.
>
>
> **W1/Q1: Calibration / SSR.** U-Cast’s overconfidence is most notable for geopotential variables, while for temperature, humidity, and wind variables, U-Cast's calibration is roughly in line with the other baselines; it’s notably better than NeuralGCM ENS at short lead times. More importantly, we don't see this as a fundamental limitation of our framework. It can be addressed through initial-condition (IC) perturbations, commonly used in the literature but orthogonal to our contributions. Notably, GenCast itself relies on such sophisticated perturbations; without them, it exhibits similar undercalibration to U-Cast. See Section F.11 and Figure F30 in the GenCast paper (https://arxiv.org/abs/2312.15796), where the non-perturbed version ("GenCast No EDA init") exhibits undercalibration at a similar scale to U-Cast's. The GenCast authors note that these perturbations have "very little effect" on CRPS, with their "primary impact on the dispersion of the ensemble".
> **[New]** Additionally, we trained a deep ensemble (DE) variant of U-Cast—uniquely enabled by our low Stage 2 training cost—by repeating Stage 2 4× with different seeds. The DE variant consistently achieves SSRs above 0.85 for all variables/lead times, and above 0.9 for non-geopotential variables—a dramatic improvement over SSRs previously reported for MC Dropout weather ensembles (e.g., Garg et al. (2022: SSR 0.34–0.4). U-Cast DE also improves CRPS scores by up to 4%. For applications where near-perfect calibration is critical, we are confident that IC perturbations could close the remaining gap.
> Regarding the dropout rate: our ablation (Fig. 4) shows minor impact on CRPS/SSR, with lower rates even slightly improving calibration. Analyzing this relationship deeper is a complementary future direction.
>
>
> **W2: Polar artifacts.** As with Reviewer XGCf: this is a fair point that we address in our Limitations. A HEALPix U-Net, as we suggest, could resolve this while preserving our framework's simplicity and efficiency. We don't view it as a fundamental limitation.
>
>
> **W3/Q2: No precipitation.** We intentionally omitted precipitation because it is a diagnostic variable that is not well represented in ERA5 reanalysis data (see Section 6.1 of the WeatherBench 2 paper). We see limited value in reporting scores against unreliable targets. Moreover, precipitation is such a unique variable, requiring bespoke evaluation and analysis, that previous works dedicate entire separate papers to it (e.g., the recent NeuralGCM precipitation study in Science Advances, Yuval et al. 2026).
>
>
> **W4: Limited novelty.** We emphasize that the specific combination of components, carefully chosen for simplicity, efficiency, and empirical performance, is non-trivial. The curriculum is the most novel element, and we now support it with a dedicated ablation (see our response to Reviewer Q4Cq, W2). We would also welcome pointers to prior work on "deterministic pre-training followed by probabilistic fine-tuning" in other domains, as we'd like to cite such precedents appropriately.

---

> > ### Author Rebuttal · Reviewer_SffL · 2026-04-04
> >
> > I thanks the authors for their response; the original score already reflected the paper's strengths and limitations, and I would like to keep my positive score.

---

### Official Review · Reviewer_XGCf · 2026-03-18

**Soundness:** 3
**Presentation:** 3
**Significance:** 3
**Originality:** 2
**Overall Recommendation:** 4
**Confidence:** 5

**Summary:**

The authors propose U-Cast, a U-Net based probabilistic weather forecasting model that challenges the trend of increasing architectural complexity in AI weather prediction. Instead of using complex GNN or Transformers, U-Cast employs a standard U-Net backbone trained with a two-stage curriculum: deterministic pre-training using MAE followed by probabilistic fine-tuning using CRPS. Uncertainty is modeled via Monte Carlo Dropout rather than sophisticated noise injection methods. The model uses the Muon optimizer and claims to match or exceed the performance of state-of-the-art models like GenCast at 1-degree resolution while being significantly more computationally efficient in both training and inference.

**Compliance With Llm Reviewing Policy:**

Affirmed.

**Final Justification:**

My concerns have been addressed well, I will retain the positive rating.

**Key Questions For Authors:**

1. About the GenCast baseline: Is the "1-degree GenCast" you compared against actually a fully trained, converged model optimized for that resolution? Or did you just take a checkpoint from a model designed for 0.25 degrees? If it wasn't retrained from scratch for 1 degree, the comparison might be unfair.
2. You say Muon is key for fine-tuning. Did you try pushing AdamW harder? I'm wondering if Muon is fundamentally better here, or if you just found better hyperparameters for it than for AdamW.
3. Those polar artifacts: Since "cylinder" padding isn't cutting it, have you thought about cheap geometric fixes like Healpix or coordinate features? You don't need a full GraphNet to handle the poles better.
4. What happens if you scale this to 0.25 degrees? U-Nets get heavy as the grid grows. Would your efficiency wins disappear compared to GraphNets at operational resolutions?
5. Why leave out the "Skillful joint probabilistic weather forecasting from marginals" paper (Alet et al., 2025)? That work is super relevant because they also train on marginals (using CRPS) but use learned perturbations to get state-of-the-art joint performance. Comparing your MC Dropout against their approach would really show if your "simpler" method can hold its own against the top-tier version of the same idea.

**Limitations:**

The authors frankly acknowledge the polar artifacts and the slight under-dispersion of the ensemble. They also note the degradation in long-term temperature skill. However, a significant limitation is not fully addressing the scalability to 0.25-degree resolution, which is the current gold standard for operational AI weather models.

**Strengths And Weaknesses:**

Strengths:
1. The paper provides a compelling counter-narrative to the "complexity trap" in AI weather forecasting. By demonstrating that a standard U-Net with proper training strategies can match SOTA performance, it lowers the barrier to entry for researchers without massive compute resources.
2. The proposed training curriculum (deterministic pre-training + probabilistic fine-tuning) effectively decouples the learning of atmospheric dynamics from the learning of uncertainty. This significantly reduces the computational cost since the expensive ensemble-based CRPS training is only needed for a short fine-tuning phase.
3. The use of the Muon optimizer in this domain is a novel and practical contribution, showing clear convergence speed improvements over standard AdamW, which is valuable for the community.

Weaknesses:
1. The comparison with GenCast at 1-degree resolution might optionally favor U-Cast. GenCast was designed and optimized for 0.25-degree resolution. Comparing a model natively trained at 1-degree (U-Cast) against a checkpoint of a model intended for higher resolution (or a precursor) might not fully capture the capability gap, especially regarding fine-scale features.
2. The paper admits to "systematic artifacts in the polar regions" (Appendix C.4) due to the lack of strict geometric priors (handling the sphere as a cylinder with padding). While the authors argue these are minor for global metrics, for a "frontier" model, failing to handle spherical topology correctly is a known regression compared to geometrically valid architectures like GraphCast.
3. The reliance on MC Dropout for uncertainty has known limitations, often producing under-dispersive ensembles compared to principled probabilistic methods. The authors claim it works well here, but Figure 4 shows the spread-skill ratio is often below 1, indicating overconfidence. The "simplicity" here might come at the cost of true probabilistic calibration.

---

> ### Author Rebuttal · Authors · 2026-03-31
>
> We thank Reviewer XGCf for the careful review and for recognizing that our paper "provides a compelling counter-narrative to the complexity trap." We address weaknesses and questions below.
>
> **W1/Q1: GenCast comparison fairness.** The 1° GenCast used in Section 4.2 is Google's official `GenCast 1p0deg <2019>` checkpoint: A fully trained, converged model (trained for 2 million steps using 112 TPUv5-days, >10× more training than U-Cast) on 1° ERA5 data. We refer to Appendix D.4.1 in the GenCast paper for further details. Separately, the 1.5° evaluation in Section 4.3 follows the standard WeatherBench 2 benchmark, where all models are evaluated at 1.5°. Here, the GenCast baseline is the native 0.25° model regridded to 1.5°. We argue that this resolution asymmetry actually *favors* higher-resolution models like GenCast, since they can learn finer-scale details, which is particularly important for near-surface variables where topography is a key factor. The consistency of our competitive results across both evaluation settings strengthens our claims.
>
> **W2/Q3: Polar artifacts.** The reviewer raises a fair point, and we're glad to see their HEALPix suggestion, which aligns with what we proposed in our paper: *"Future work could address these issues by operating the U-Net in a more suitable grid representation (Karlbauer et al., 2024)."* Because this can be resolved within the key properties of our framework (simplicity + efficiency), e.g., with a HEALPix U-Net, we don't see it as a fundamental limitation but rather something we're confident can be addressed in near-term follow-up work.
>
> **W3: MC Dropout calibration.** U-Cast’s overconfidence is most notable for geopotential variables, while for temperature, humidity, and wind variables, U-Cast's calibration is roughly in line with the other baselines; it’s notably better than NeuralGCM ENS at short lead times. More importantly, we don't see this as a fundamental limitation of our framework. It can be addressed through initial-condition (IC) perturbations, commonly used in the literature but orthogonal to our contributions. Notably, GenCast itself relies on such sophisticated perturbations; without them, it exhibits similar undercalibration to U-Cast. See Section F.11 and Figure F30 in the GenCast paper (https://arxiv.org/abs/2312.15796), where the non-perturbed version ("GenCast No EDA init") exhibits undercalibration at a similar scale to U-Cast's. The GenCast authors note that these perturbations have "very little effect" on CRPS, with their "primary impact on the dispersion of the ensemble".
> **[New]** Additionally, we trained a deep ensemble (DE) variant of U-Cast—uniquely enabled by our low Stage 2 training cost—by repeating Stage 2 4× with different seeds. The DE variant consistently achieves SSRs above 0.85 for all variables/lead times, and above 0.9 for non-geopotential variables—a dramatic improvement over SSRs previously reported for MC Dropout weather ensembles (e.g., Garg et al. 2022: SSR 0.34–0.4). U-Cast DE also improves CRPS scores by up to 4%. For applications where near-perfect calibration is critical, we are confident that IC perturbations could close the remaining gap.
>
> **Q2: Muon optimizer.** Beyond the ablation in Fig. 4, we extensively explored AdamW during development. While both optimizers reach comparable optima for deterministic pre-training given sufficient time, Muon provides clear benefits during probabilistic fine-tuning, where AdamW struggled to improve upon the deterministic baseline. As discussed in our reply to Q4Cq, we believe a well-tuned AdamW pipeline could match our results given enough compute, but at the cost of the training efficiency central to our contribution. We hope future work can shed light on whether Muon's benefits extend to the vision and graph transformers common in AI for Earth sciences.
>
> **Q4: Scaling to 0.25°.** This is an important direction for future work, but difficult to address imminently due to the sheer scale of 0.25° data (>30 TB storage). However, we're confident in its feasibility: (1) current 0.25° SOTA models (GenCast, FGN) are fine-tuned from 1° versions, making a similar curriculum natural for U-Cast; (2) hierarchical cascading from lower to higher resolutions is well-established in the image generation literature using standard U-Nets (e.g., Ho et al., "Cascaded Diffusion Models for High Fidelity Image Generation", JMLR 2022).
>
> **Q5: Alet et al.** FGN (Alet et al., 2025) is one of our most-cited references, as we agree it is highly relevant. We cannot directly compare against it because it's closed-source. However, we include a crucial ablation in Fig. 4, in which we apply the same "learned perturbations" idea from FGN (adaLN noise injection) within our U-Cast framework, and find that it deteriorates CRPS by up to 3–4% while improving SSR. We believe this demonstrates that our simpler MC Dropout approach holds up well against more complex SOTA stochasticity mechanisms.

---

> > ### Author Rebuttal · Reviewer_XGCf · 2026-04-04
> >
> > Thanks for the authors' detailed response.  Since my concerns have been addressed well, I will retain the positive rating.

---

### Decision · Program_Chairs · 2026-04-30

**Decision:**

Accept (regular)

**Comment:**

This paper proposes U-Cast, a U-Net-based probabilistic weather forecasting model using MC Dropout and a two-stage training curriculum (deterministic pre-training and probabilistic fine-tuning) with the Muon optimizer. Its main strengths include matching SOTA performance at dramatically lower compute cost, an effective curriculum that decouples dynamics learning from uncertainty estimation, and honest empirical evaluation. After the rebuttal, concerns related to ensemble calibration, GenCast comparison fairness, curriculum ablation, and the Muon optimizer's role were satisfactorily resolved.